EMBO
Molecular Medicine

# Disease-modifying effects of ganglioside GM1 in Huntington's disease models

Melanie Alpaugh[1,2], Danny Galleguillos[1,2], Juan Forero[2,3], Luis Carlos Morales[1], Sebastian W Lackey[1], Preeti Kar[1], Alba Di Pardo[1,†], Andrew Holt[1], Bradley J Kerr[2,4], Kathryn G Todd[2,5], Glen B Baker[2,5], Karim Fouad[2,3] & Simonetta Sipione[1,2,*] 

## Abstract

Huntington's disease (HD) is a progressive neurodegenerative disorder characterized by motor, cognitive and psychiatric problems. Previous studies indicated that levels of brain gangliosides are lower than normal in HD models and that administration of exogenous ganglioside GM1 corrects motor dysfunction in the YAC128 mouse model of HD. In this study, we provide evidence that intraventricular administration of GM1 has profound disease-modifying effects across HD mouse models with different genetic background. GM1 administration results in decreased levels of mutant huntingtin, the protein that causes HD, and in a wide array of beneficial effects that include changes in levels of DARPP32, ferritin, Iba1 and GFAP, modulation of dopamine and serotonin metabolism, and restoration of normal levels of glutamate, GABA, L-Ser and D-Ser. Treatment with GM1 slows down neurodegeneration, white matter atrophy and body weight loss in R6/2 mice. Motor functions are significantly improved in R6/2 mice and restored to normal in Q140 mice, including gait abnormalities that are often resistant to treatments. Psychiatric-like and cognitive dysfunctions are also ameliorated by GM1 administration in Q140 and YAC128 mice. The widespread benefits of GM1 administration, at molecular, cellular and behavioural levels, indicate that this ganglioside has strong therapeutic and disease-modifying potential in HD.

**Keywords** behaviour; gangliosides; HD mouse models; huntingtin; neuroprotection

**Subject Categories** Neuroscience; Pharmacology & Drug Discovery

## Introduction

Huntington's disease (HD) is a dominantly inherited neurodegenerative disorder caused by the pathological expansion of a trinucleotide (CAG) repeat in the gene that codes for huntingtin (HTT) (Group THsDCR 1993). This mutation results in an abnormally long polyglutamine stretch at the N-terminus of the mutant HTT (mHTT) protein, which in turn causes mHTT to misfold into toxic species and to aggregate. A plethora of molecular, cellular and network dysfunctions ensue, eventually leading to neuronal death (Imarisio *et al*, 2008). Cerebral cortex and corpus striatum are the most affected brain regions (Vonsattel *et al*, 2011), while more subtle pathological changes occur in other brain areas (Petersen & Bjorkqvist, 2006; Aziz *et al*, 2009; Vonsattel *et al*, 2011).

Motor dysfunction is the hallmark of HD and defines disease onset (Roos, 2010). However, cognitive and psychiatric problems often precede the appearance of motor symptoms and are frequently the most distressing for patients and their families (Paulsen *et al*, 2008; Roos, 2010).

To date, there is no cure or disease-modifying therapy for HD. Clinical management of HD symptoms is possible to a certain extent with the use of tetrabenazine to reduce chorea, and with traditional antidepressant and anti-psychotic drugs (Ross & Tabrizi, 2011). However, the use of these symptomatic treatments is often limited by their potential side effects (Rosenblatt, 2007; Killoran & Biglan, 2014), and the overall efficacy of antidepressants in HD patients is controversial (Moulton *et al*, 2014). Furthermore, none of these treatments target the underlying causes of dysfunction nor are able to slow down HD progression.

In previous studies, we showed that the synthesis of gangliosides—sialic acid-containing glycosphingolipids—is affected in cellular and animal models of HD (Desplats *et al*, 2007; Denny *et al*, 2010; Maglione *et al*, 2010), resulting in lower levels of ganglioside GM1 and, to a lesser extent, other major brain gangliosides (Maglione *et al*, 2010).

1   Department of Pharmacology, University of Alberta, Edmonton, AB, Canada
2   Neuroscience and Mental Health Institute, University of Alberta, Edmonton, AB, Canada
3   Faculty of Rehabilitation Medicine, University of Alberta, Edmonton, AB, Canada
4   Department of Anesthesiology and Pain Medicine, University of Alberta, Edmonton, AB, Canada
5   Department of Psychiatry, University of Alberta, Edmonton, AB, Canada
    *Corresponding author. Tel: +1 780 492 5885; E-mail: ssipione@ualberta.ca
    †Present address: Center for Neurogenetics and Rare Diseases, IRCCS Neuromed, Pozzilli, Italy

Gangliosides perform a plethora of modulatory functions in cell signalling, cell–cell interactions and calcium homeostasis (Sonnino et al, 2007; Posse de Chaves & Sipione, 2010; Ledeen & Wu, 2015). Their importance in the central nervous system is underscored by the fact that knockout mouse models that lack complex gangliosides undergo neurodegeneration (Proia, 2003) and display motor impairment (Chiavegatto et al, 2000), depression-like behaviour (Wang et al, 2014) and learning and memory deficits (Sha et al, 2014). This suggests that reduced GM1 levels in HD may contribute to disease pathogenesis and/or progression. In support of this hypothesis, we showed that administration of exogenous GM1 decreases HD cell susceptibility to apoptosis in vitro (Maglione et al, 2010) and corrects motor dysfunction in the YAC128 mouse model (Di Pardo et al, 2012). However, whether the treatment was able to ameliorate other important aspects of the disease—including non-motor symptoms—and to attenuate the underlying molecular dysfunctions and neurodegeneration remained to be investigated.

In this study, we used three different—and for many aspects complementary—HD mouse models (R6/2, Q140 and YAC128 mice) (Mangiarini et al, 1996; Menalled et al, 2003; Slow et al, 2003) to show that chronic intraventricular infusion of GM1 has profound disease-modifying effects that include a reduction in brain levels of toxic mHTT, restoration of normal brain concentrations of specific neurotransmitters, as well as decreased neurodegeneration and white matter atrophy, among others. These changes are accompanied by improvement or even restoration of motor function and by correction of behavioural abnormalities related to depression, anxiety and cognition.

# Results

## Intracerebroventricular infusion of GM1 slows down neurodegeneration, improves neuropathology and decreases body weight loss in R6/2 mice

To determine the effects of GM1 on HD brain neuropathology, we performed chronic intracerebral infusion of this ganglioside in R6/2 mice (Appendix Fig S1). Compared to other models used in this study, R6/2 mice present with an accelerated disease phenotype and develop widespread neurodegeneration from a young age (Mangiarini et al, 1996). Treatment with GM1 significantly attenuated striatal atrophy (effect of treatment: $F_{1,40} = 152.4$, $P < 0.0001$—see also Appendix Table S1 for P-values relative to pairwise comparisons; Fig 1A) and loss of striatal neurons (effect of treatment: $F_{1,35} = 21.0$, $P < 0.001$) compared to vehicle (artificial cerebrospinal fluid, CSF)-treated R6/2 mice (Fig 1B). Brain volume (between bregma 1.98 and −2.3; Fig 1C) and total brain weight (Fig 1D) were also significantly higher in R6/2 mice treated with GM1 compared to vehicle-treated mice (effect of treatment: $F_{1,41} = 14.33$, $P < 0.001$; effect on brain volume: $P = 0.0092$). While brain weight decreased progressively in R6/2 mice from 6 to 10 weeks of age (effect of age: $F_{3,33} = 10.86$, $P < 0.0001$), administration of GM1 for 4 weeks significantly slowed down this process, maintaining brain weight at levels observed at 8 weeks of age without treatment ($P = 0.99$). Altogether, these data suggest that treatment with GM1 significantly slowed down neurodegeneration in R6/2 mice.

GM1 treatment resulted also in decreased atrophy of the corpus callosum (Fig 1E) (interaction: $F_{1,42} = 4.26$, $P = 0.0451$; GM1-treated R6/2 vs. WT: $P = 0.9553$) and of cortico-striatal white matter tracts (Fig 1F) (effect of genotype: $F_{1,43} = 54.01$, $P < 0.0001$; R6/2 CSF vs. GM1: $P = 0.0473$) in R6/2 mice that received the ganglioside compared to CSF-treated R6/2 mice. Changes in these white matter structures predate neuronal death and the onset of motor symptoms in HD patients (Paulsen et al, 2006; Jech et al, 2007; Tabrizi et al, 2011, 2012), and correlate with cognitive (Crawford et al, 2013; Novak et al, 2014; Matsui et al, 2015) and cortico-striatal functional deficits (Sapp et al, 1999; Wolf et al, 2008; Douaud et al, 2009), respectively.

Dysregulation of iron metabolism and increased ferritin expression are pathological changes that occur in HD patients (Bartzokis et al, 2007b) and R6/2 mice (Simmons et al, 2007) and are likely linked to oxidative stress (Chen et al, 2013a) and dystrophic microglia (Simmons et al, 2007). Upon treatment with GM1, ferritin levels in the brain of R6/2 mice were significantly decreased (interaction: $F_{1,14} = 14.49$, $P = 0.0019$) (Fig 1G).

Survival and body weight are frequently used as indices of treatment efficacy in R6/2 mice (Li et al, 2005). Body weight loss was significantly attenuated by administration of GM1 (interaction: $F_{1,65} = 6.43$, $P < 0.01$; Fig 1H). This was not due to increased food intake, which was not significantly different among groups (Appendix Fig S2). Furthermore, GM1 treatment did not affect the weight of WT mice ($P = 0.553$), suggesting that weight increase in R6/2 mice is the result of overall improved health conditions. R6/2 mice receiving GM1 also showed a trend towards increased lifespan, although these results were not statistically significant ($P = 0.10$; Fig 1I).

## GM1 attenuates pathological cellular and molecular changes in HD mice

Because of the increasingly recognized role of non-neuronal cell populations in HD pathogenesis (Lobsiger & Cleveland, 2007; Ehrlich, 2012), we next measured the effects of GM1 on astrocytic and microglial markers that are affected by neuroinflammation and in HD (Singhrao et al, 1999; Laurine et al, 2003; Stack et al, 2006; Dalrymple et al, 2007; Giampa et al, 2010; Politis et al, 2011; Baune, 2015; Olejniczak et al, 2015). Immunohistochemistry (Fig 2A and B) and protein immunoblotting (Fig 2C) showed similar levels of glial fibrillary acid protein (GFAP) immunoreactivity in the striatum of WT and R6/2 mice, regardless of treatment. In cortical sections from both CSF- and GM1-treated R6/2 mice, the area covered by GFAP immunoreactivity was slightly higher than in WT littermates (effect of genotype, $F_{1,22} = 4.32$, $P = 0.049$; Fig 2B). In spite of this, GFAP protein expression measured by immunoblotting of cortical lysates was significantly decreased in CSF-treated R6/2 mice compared to WT (effect of genotype: $F_{1,24} = 13$, $P = 0.0014$; Fig 2C). This apparent discrepancy between immunohistochemistry and protein immunoblotting data (i.e. decreased protein expression of GFAP in spite of a larger area covered by GFAP$^+$ astrocytes in cortical tissue) might reflect the coexistence of quantitative as well as qualitative changes in HD astroglia. Administration of GM1 restored GFAP protein expression to WT levels ($P = 0.028$; Fig 2C).

Similar effects of GM1 were observed on the microglial marker Iba1. Although the number of Iba1$^+$ cells was similar in the cortex

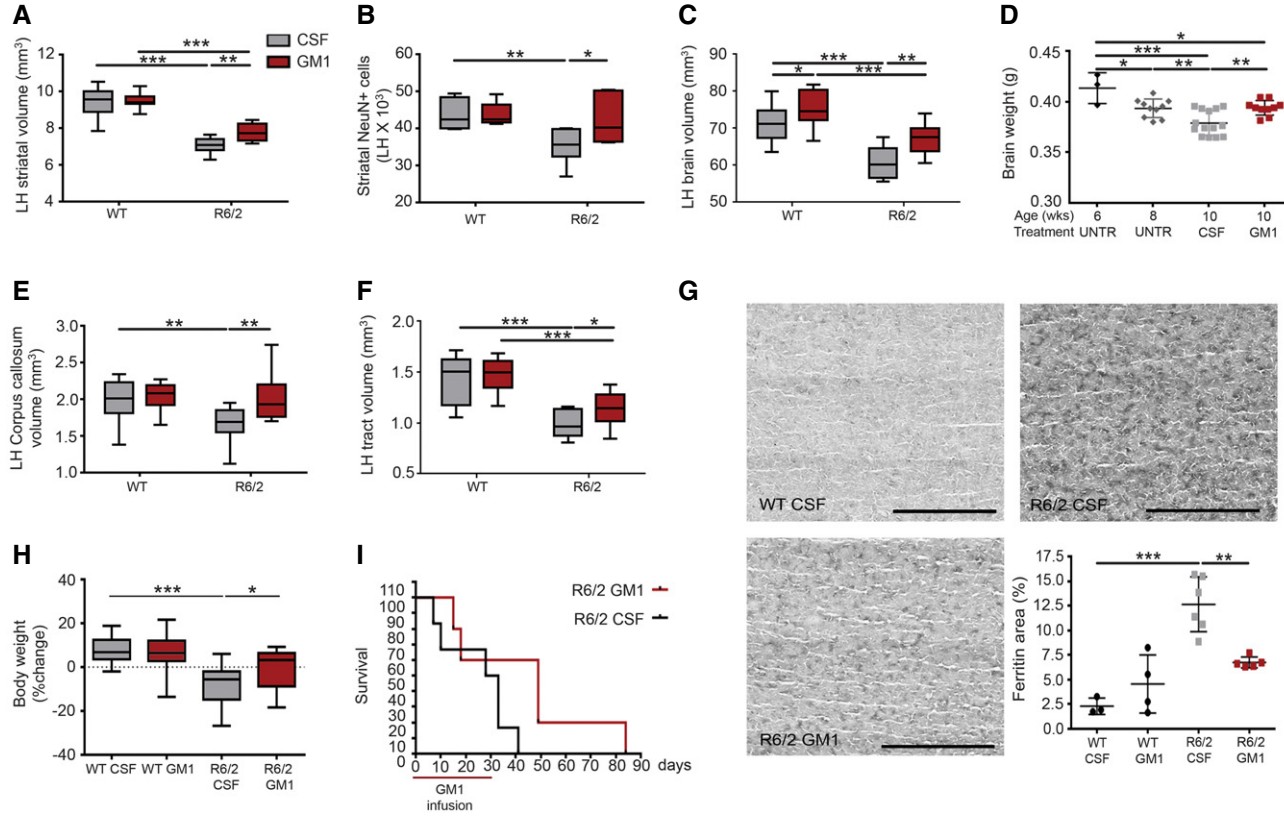

**Figure 1. GM1 decreases neuropathology and weight loss in R6/2 mice.**

Six-week-old mice were infused with GM1 or artificial cerebro-spinal fluid (CSF, vehicle) for 28 days. Analysis was performed at the end of treatment (10 weeks of age).

A    Striatal volume in the brain left hemisphere (LH, contralateral to infusion site). *N* = 13 WT CSF, 11 WT GM1, 11 R6/2 CSF, 9 R6/2 GM1.
B    Number of neurons (NeuN⁺ cells) in the LH striatum between bregma 0.02 mm and −2.3 mm. *N* = 8 WT CSF, 5 WT GM1, 7 R6/2 CSF, 6 R6/2 GM1.
C    Volume of the brain (LH) from bregma 2.1 mm to −2.3 mm. *N* = 13 WT CSF, 11 WT GM1, 11 R6/2 CSF, 9 R6/2 GM1.
D    Time course of brain weight loss in R6/2 mice. *N* = 3 6-week R6/2, 10 8-week R6/2, 14 10-week R6/2 CSF and 10 10-week R6/2 GM1.
E    Corpus callosum volume (LH) between bregma 2.1 mm and 0.02 mm. *N* = 13 WT CSF, 11 WT GM1, 11 R6/2 CSF, 9 R6/2 GM1.
F    Total white matter tract volume in the striatum, from 0.02 mm to bregma to −1.06 mm. *N* = 13 WT CSF, 11 WT GM1, 11 R6/2 CSF, 9 R6/2 GM1.
G    Representative microscopy images of the striatum after immunostaining with anti-ferritin antibodies. Scale bars are 0.62 mm in length. Quantification of the immunoreactive area is shown in the graph. Eight serial sections were analysed and averaged for each mouse. *N* = 3 WT CSF, 4 WT GM1, 6 R6/2 CSF, 5 R6/2 GM1.
H    Percent change in body weight at day 21 of treatment compared to baseline (day 0). *N* = 23 WT CSF, 21 WT GM1, 14 R6/2 CSF, 11 R6/2 GM1.
I    Survival curve for R6/2 mice treated with CSF (*N* = 6) or GM1 (*N* = 5). *X*-axis shows days after the beginning of GM1 treatment. The horizontal red line indicates the duration of GM1 treatment.

Data information: Box-and-whisker plots show median, maximum and minimum values. Two-way ANOVA with Holm–Sidak post-test was used in (A–C and E–H); one-way ANOVA with Tukey's post-test in (D); log-rank analysis was used in (I). *$P < 0.05$, **$P < 0.01$, ***$P < 0.001$.

of R6/2 and WT mice ($P = 0.69$) and only slightly decreased in R6/2 striatum ($P = 0.03$; Fig 2E), unexpectedly Iba1 protein expression was lower than normal in cortical tissue from R6/2 mice (effect of genotype: $F_{1,23} = 20.62$, $P = 0.001$) (Fig 2F). A similar trend was also observed in striatal tissue from R6/2 mice, although here statistical significance was not reached ($P = 0.10$). GM1 treatment had genotype-dependent effects, decreasing Iba1 protein levels in the striatum of WT animals, but increasing them to control levels in R6/2 mice (interaction: $F_{1,24} = 13.32$, $P = 0.001$; Fig 2F). Like in the case of GFAP expression, our data suggest qualitative differences in the molecular signature of R6/2 microglia that were at least in part mitigated by administration of GM1. In support of this conclusion, expression of the gene encoding the transcription factor Spi-1, a master regulator of microglia biogenesis and function (Heinz *et al*, 2010; Kierdorf *et al*, 2013), was lower in vehicle-treated R6/2 mice

than in WT animals, and restored to WT levels by GM1 (Fig 2G). Our data also suggest the absence of an overt neuroinflammatory microglia phenotype in our R6/2 mice and experimental conditions. In line with these data, brain levels of major inflammatory cytokines were similar (and low) across genotype and treatment groups (Appendix Fig S3).

Next, we determined the effects of GM1 on DARPP32, a key regulator of dopamine (DA) signalling and striatum output pathways (Greengard *et al*, 1999; Svenningsson *et al*, 2004). Downregulation—as well as decreased phosphorylation—of DARPP32 marks early dysfunction of HD medium spiny neurons (Bibb *et al*, 2000). For the analysis of DARPP32 levels (as for GFAP analysis), we only used male mice, to avoid potential confounding effects of the oestrous cycle in female animals (Bode *et al*, 2008; Hajos, 2008). Chronic treatment with GM1 for 42 days increased expression of

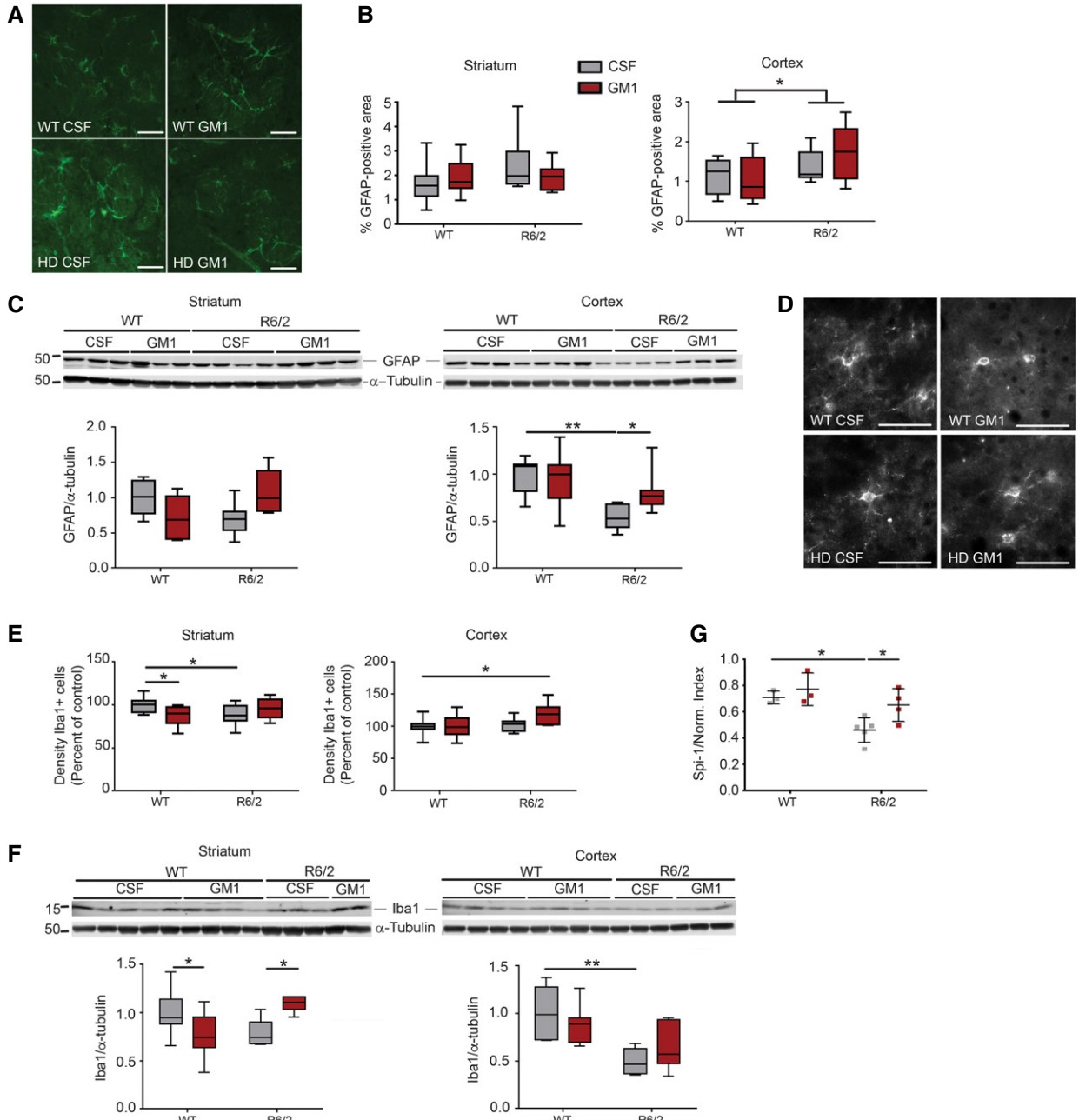

**Figure 2. Effects of GM1 on astroglial and microglial markers.**

R6/2 and WT mice were treated with artificial cerebro-spinal fluid (CSF, vehicle) or GM1 for 28 days.

A   Representative brain section staining with anti-GFAP antibodies. Areas shown are in the corpus striatum. Scale bars are 50 μm in length.

B   Graphs show the quantification of GFAP-immunoreactive area in micrographs of coronal serial sections. For each mouse, eight serial sections were analysed and averaged. $N$ = 11 WT CSF, 9 WT GM1, 10 R6/2 CSF, 8 R6/2 GM1.

C   GFAP protein expression in tissue lysates. Representative immunoblots and densitometric analysis are shown. $N$ = 7 WT CSF, 7 WT GM1, 7 R6/2 CSF, 7 R6/2 GM1. The immunoblot showing α-tubulin in the cortex is the same as for the cortex in (F), since GFAP and Iba1 were run in the same gel.

D, E Representative micrographs (D) (from the striatum) and quantification (E) of Iba1[+] cell density in the cortex and striatum. For each mouse, eight serial sections were analysed and averaged. Scale bars are 50 μm in length. $N$ = 11 WT CSF, 10 WT GM1, 10 R6/2 CSF, 9 R6/2 GM1.

F   Iba1 protein expression in tissue lysates. Representative immunoblots and densitometric analysis are shown. The immunoblot showing α-tubulin in the cortex is the same as for the cortex in (C), since GFAP and Iba1 were run in the same gel. $N$ = 7 WT CSF, 7 WT GM1, 6 R6/2 CSF, 7 R6/2 GM1.

G   *Spi-1* gene expression in the striatum, analysed by qPCR and normalized over the geometric mean of three stably expressed reference genes. $N$ = 3 WT CSF, 3 WT GM1, 5 R6/2 CSF, 4 R6/2 GM1.

Data information: Box-and-whisker plots show median, maximum and minimum values. Two-way ANOVA with Holm–Sidak post-test. *$P < 0.05$; **$P < 0.01$.
Source data are available online for this figure.

DARPP32 and pDARPP32 to WT levels in heterozygous Q7/140 mice (for DARPP32: $F_{4,65} = 23.88$, $P < 0.0001$; Q7/140 CSF vs. GM1 $P < 0.001$; Q7/7 vs. Q7/140 GM1, $P = 0.71$. For pDARPP32: $F_{4,65} = 23.2$, $P < 0.0001$; Q7/140 CSF vs. GM1 $P < 0.001$; Q7/7 vs. Q7/140 GM1, $P = 0.93$) (Fig EV1). This suggests that GM1 has overall beneficial effects on HD medium spiny neurons signalling and functionality. These specific effects, however, did not extend to homozygous Q140/140 mice, for reasons that are currently unknown (see Discussion).

## Mutant HTT levels are reduced by administration of GM1

Mutant HTT is arguably a primary therapeutic target in HD. GM1 treatment caused a reduction of mHTT protein levels in the striatum of both Q7/140 and Q140/140 mice (effect of treatment: $F_{1,11} = 8.798$, $P = 0.012$; HD CSF vs. HD GM1 $P = 0.023$) after 42 days of treatment (Fig 3A; but not yet after 28 days—data not shown). Since *Htt* mRNA expression was not affected by GM1 ($P = 0.45$; Fig 3B), decreased mHTT levels were likely due to increased protein clearance. Wild-type HTT was not significantly affected by the treatment ($P = 0.95$).

GM1 also decreased accumulation of SDS-insoluble aggregates of mHTT in the striatum of Q140/140 mice ($P = 0.005$; Fig 3C). A similar effect was observed in the cortex ($P = 0.001$), although not in the striatum, of GM1-treated R6/2 mice (Fig 3D). In heterozygous Q7/140 mice, a low amount of SDS-insoluble aggregates and high inter-animal variability likely prevented us from detecting any significant effect of GM1 treatment (data not shown).

## Treatment with GM1 improves motor performance in R6/2 and Q140 mice

The beneficial action of GM1 on HD neuropathology in R6/2 and Q140 mice correlated with amelioration of motor dysfunction in both animal models (Figs 4 and 5), in line with our previous observations in YAC128 mice (Di Pardo *et al*, 2012).

In the horizontal ladder test, a measure of skilled motor control (Metz & Whishaw, 2002), GM1 significantly improved performance of R6/2 mice at each time point from the beginning of treatment ($P \leq 0.01$, day 7 95% CI: $-19.4$ to $-0.1$; day 11 95% CI: $-19.9$ to $-1.6$; day 15 95% CI: $-21.1$ to $-2.5$, day 21 95% CI: $-23.9$ to $-2.7$; Fig 4A). In the same test, we also observed a significant improvement in Q140 mice after treatment with GM1 past 28 days (effect of treatment: $F_{2,73} = 21.19$, $P \leq 0.0001$; Fig 4B). For this test, data from male Q7/140 and Q140/140 mice were combined as their performance was similar (Appendix Table S2). Female Q7/140 and Q140/140 were not significantly different from WT ($P > 0.99$). Compared to vehicle, GM1 administration also prevented a progressive decline of ambulatory and exploratory activity in R6/2 mice (Fig 4C; day 7 of treatment, 95% CI: 79.1–673.2, $P \leq 0.001$; day 11, 95% CI: 68.3–828.2, $P \leq 0.001$; day 15, 95% CI: 23.8–1017.0 $P \leq 0.01$; day 21, 95% CI: 85.6–1171.6, $P \leq 0.01$). Similar effects were observed in Q140 mice (treatment: $F_{2,76} = 15.58$, $P < 0.0001$), where GM1 increased performance to wild-type levels ($P < 0.01$ compared to vehicle-treated Q140; $P = 0.2$ compared to Q7/7; Fig 4D), and restored normal rearing behaviour ($P = 0.0005$ compared to vehicle-treated Q140; $P > 0.99$ compared to Q7/7; Fig 4E). For both

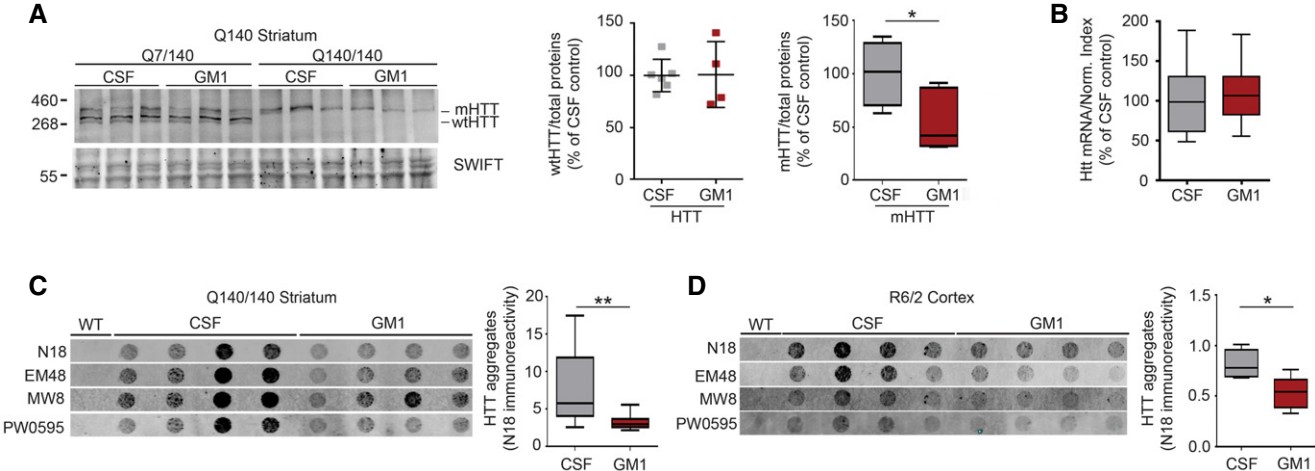

**Figure 3. Mutant HTT protein levels are reduced by administration of GM1.**

A Representative immunoblots and densitometric analysis of wtHTT and mHTT in striatal tissue lysates from Q140 mice after 42 days of treatment with artificial cerebro-spinal fluid (CSF, vehicle) or GM1. Swift™ total protein staining was used as loading control and for data normalization. Homozygous and heterozygous Q140 mice were pooled for the analysis of mHTT as GM1 had similar effects in both genotypes. For the box-and-whisker plot: N = 9 Q140 CSF and 7 Q140 GM1.

B Analysis of total HTT mRNA expression in CSF- and GM1-treated Q140 mice. Data from heterozygous Q7/140 and homozygous Q140/140 mice were similar and were combined. N = 18 Q140 CSF, 17 Q140 GM1.

C, D Filter-trap assay for mHTT insoluble aggregates in tissue lysates. Representative immunoblots and densitometric analysis are shown. N = 12 Q140/140 CSF, 11 Q140/140 GM1, five R6/2 CSF, six R6/2 GM1. SDS-insoluble mHTT aggregates were detected with the indicated anti-HTT antibodies. Only the densitometric analysis for N18 immunoreactivity is shown.

Data information: Box-and-whisker plots show median, maximum and minimum values. Two-tailed Student's *t*-test (A, B) and Mann–Whitney test (C, D). *$P < 0.05$, **$P < 0.01$.

 

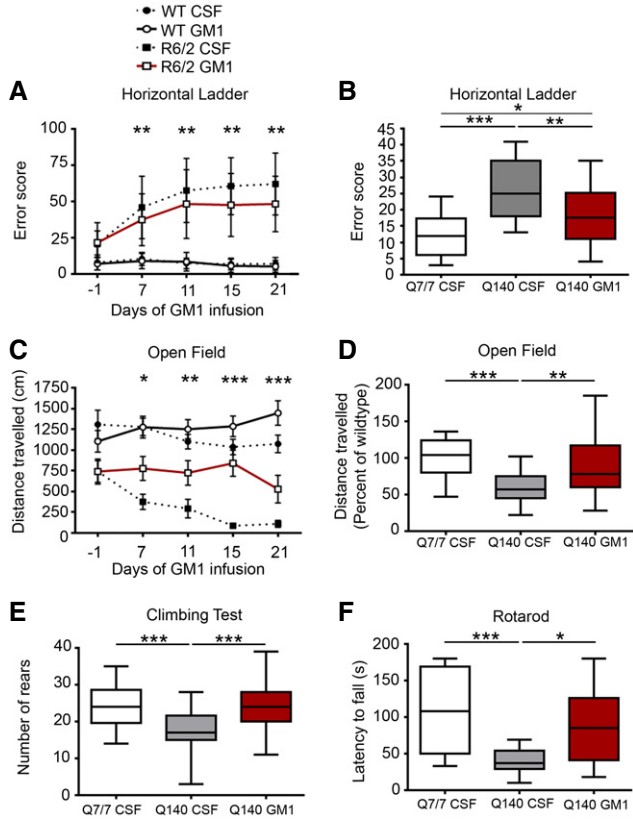

**Figure 4.  GM1 improves motor behaviour in R6/2 and Q140 mice.**

Motor testing was performed during treatment with cerebro-spinal fluid (CSF, vehicle) or GM1, between day 7 and day 21 of treatment for R6/2 mice, and after day 28 of treatment for Q140 mice.

A, B   Horizontal ladder test. Total error score from five consecutive passes is shown. N = 24 WT CSF, 21 WT GM1, 20 R6/2 CSF, 18 R6/2 GM1; N = 26 Q7/7 CSF, 23 Q140 CSF, 26 Q140 GM1.

C, D   Open field activity test. The distance travelled during 5-min-long sessions is reported. For Q140 mice, the distance travelled relative to Q7/7 is shown. N = 23 WT CSF, 21 WT GM1, 20 R6/2 CSF, 17 R6/2 GM1; N = 27 Q7/7 CSF, 30 Q140 CSF, 22 Q140 GM1.

E   Climbing test. Number of rears performed in 5 min of placement in a wire mesh container. N = 25 Q7/7 CSF, 29 Q140 CSF, 27 Q140 GM1.

F   Fixed-speed (12 RPM) rotarod test. Latency to fall is the average of three consecutive trials for each animal. N = 14 Q7/7 CSF, 15 Q140 CSF, 9 Q140 GM1, all females.

Data information: Box-and-whisker plots show median, maximum and minimum values. Statistical analysis in (A, C) was performed using a linear mixed effect regression model with 95% confidence intervals calculated at each time point; asterisks indicate statistically significant differences between HD CSF and HD GM1. For all other data, one-way ANOVA with Bonferroni correction was used. *$P < 0.05$, **$P < 0.01$, ***$P < 0.001$.

tests, where we did not detect differences in performance between males and females and between Q7/140 and Q140/140 mice (Appendix Table S2), data from these groups were pooled and analysed together.

To determine the effects of GM1 on balance and motor coordination, we tested mice on the rotarod. Treatment with GM1 improved the performance of Q140 mice to Q7/7 levels (effect of treatment: $F_{2,36} = 8.992$, $P = 0.0007$; Q140 GM1 vs. Q7/7: $P > 0.9$; Fig 4F). Interestingly, only female Q140 mice (both Q7/140 and Q140/140),

but not males, displayed impaired behaviour in this test (effect of sex: $F_{1,56} = 6.7$, $P = 0.01$); therefore, data shown in Fig 4F include females only. In a previous study (Hickey *et al*, 2012), male and female Q140 mice were shown to perform similarly in the rotarod test, at least up to 4.5 months of age. Since our mice were 7.5 months old at the time of testing on the rotarod, our data suggest that Q140 females might develop a progressive motor impairment on this test as they age.

Gait analysis provides an additional sensitive method to assess motor impairment and basal ganglia deficits (Amende *et al*, 2005; Vandeputte *et al*, 2010; Abada *et al*, 2013). Gait abnormalities are well documented in HD patients (Delval *et al*, 2008a,b; Kegelmeyer *et al*, 2014; Casaca-Carreira *et al*, 2015) and are often resistant to treatments that improve other motor symptoms (Ferrara *et al*, 2012; Kegelmeyer *et al*, 2014). Q140 mice displayed several gait abnormalities compared to Q7/7 littermates (Fig 5). The height of the iliac crest during spontaneous walking was normal in female Q140 mice, but significantly decreased in vehicle-treated Q140 male mice compared to Q7/7 male mice ($P = 0.007$) (interaction between group and sex: $P = 0.048$), indicating a decline in weight support. GM1 treatment restored the height of the iliac crest to wild-type levels ($P = 0.007$ compared to CSF-treated Q140; $P = 0.39$ compared to Q7/7; Fig 5A and B). The average stride duration, a measure of gait speed also affected in HD patients (Thaut *et al*, 1999), was significantly longer in both male and female Q140 mice compared to Q7/7 littermates ($P = 0.007$), and corrected to normal by GM1 (Q140 GM1 vs. Q140 CSF: $P = 0.014$; Q140 GM1 vs. Q7/7 CSF: $P = 0.719$; Fig 5C). The stance-to-stride ratio, where a longer stance phase duration is indicative of reduced weight support (American Physiological Society, 1996), was significantly higher in CSF-treated Q140 mice compared to CSF-treated Q7/7 ($P = 0.002$) but restored to normal in GM1-treated Q140 mice (Fig 5E). The coupling between limbs, a measure of motor coordination and synchrony during walking, was established based on the gait diagram illustrated in Fig 5D (Leblond *et al*, 2003). Although some gait measurements were affected by the position (right or left) of the infusion pump implanted on the back of the animals (interaction between group and pump position: $P = 0.009$)—in which case data for right or left position of the pump were shown separately in Fig 5E and F—phase values in CSF-treated Q140 mice were clearly different from those in CSF-treated Q7/7 mice, and treatment with GM1 restored overall coupling to what was measured in Q7/7 mice (Fig 5E and F). Altogether, our data suggest a strong restorative effect of GM1 on motor function in HD animals, especially in Q140 mice.

## Treatment with GM1 decreases anxiety- and depression-related behaviour in HD mice

Although motor problems are the hallmark of HD and define disease onset, neuropsychiatric changes occur in nearly all HD patients and are frequently the most distressing for patients and their families (Paulsen *et al*, 2001a, 2008; Roos, 2010).

To assess the effect of GM1 treatment on non-motor behaviour, we used YAC128 and Q140 mice, which differ from each other for genetic background and strain, and where cognitive and psychiatric-like dysfunctions have been well characterized (Lione *et al*, 1999; Pouladi *et al*, 2009; Pla *et al*, 2014). R6/2 mice were only used in the spontaneous defecation test, as their profound motor

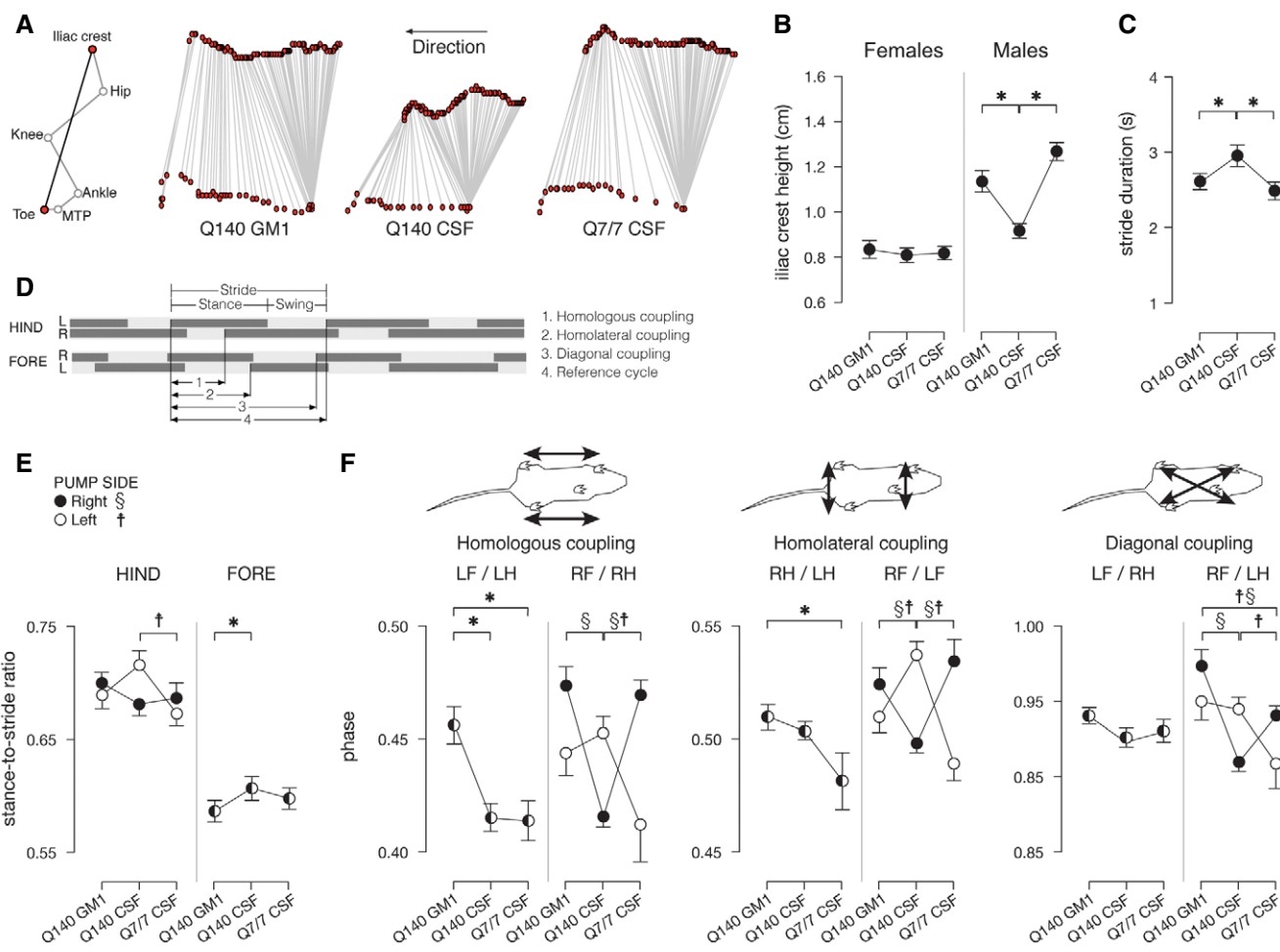

**Figure 5.  GM1 corrects gait abnormalities in Q140 mice.**

A    Representative stick diagram decompositions (5 ms between sticks) of the left iliac crest and toe motion for a Q7/7 CSF mouse, a Q140 CSF mouse and a Q140 GM1 mouse during walking on a walkway.

B    Mean (± s.e.m.) values for iliac crest height. *N* = 9 Q7/7 CSF (five males, four females), 17 Q140 CSF (seven males, 10 females), 18 Q140 GM1 (eight males, 10 females).

C    Mean (± s.e.m.) values for stride duration. *N* = 9 Q7/7 CSF (five males, four females), 17 Q140 CSF (seven males, 10 females), 18 Q140 GM1 (eight males, 10 females).

D–F    Footfall diagrams obtained from video analysis of locomotion were used to calculate: (E) stance-to-stride ratios and (F) coupling phase values (homologous LF/LH and RF/RH, homolateral RH/LH and RF/LF and diagonal LF/RH and RF/LH). Mean (± s.e.m.) values. *N* = 9 Q7/7 CSF (five males, four females), 17 Q140 CSF (seven males, 10 females), 18 Q140 GM1 (eight males, 10 females).

Data information: Statistics were performed using separate three-factor ANOVAs (GROUP [3] × SEX [2] × PUMP [2]) to determine whether there was an effect of the sex (SEX), the side at which the pump hang (PUMP) and the treatment group (GROUP) on the different kinematic measures. Significant differences (*P* < 0.05) are indicated by asterisks (*) for main effects, and by section signs (§, pump hanging on the right) and daggers (†, pump hanging on the left) for interaction effects.

impairment would have confounded results obtained with other tests of anxiety and depression.

In the elevated plus maze, vehicle-treated YAC128 mice spent significantly more time in the closed arms compared to wild-type (WT) littermates (Fig 6A; effect of genotype: $F_{1,57}$ = 5.7, $P$ < 0.05). This behaviour was not due to differences in overall exploratory activity (Appendix Fig S4A), but rather to increased anxiety, since it could be corrected by acute administration of an anxiolytic drug (adinazolam, 2.5 mg/kg; Appendix Fig S4B; Owens *et al*, 1989). Treatment with GM1 increased the time YAC128 mice spent in light arms (interaction: $F_{1,57}$ = 4.31, $P$ < 0.05; Fig 6A), suggesting that it exerted an anxiolytic effect.

In the novelty-suppressed feeding test, which is based on a different anxiogenic/depressive paradigm than the elevated plus maze

(Gross *et al*, 2000), vehicle-treated YAC128 mice took more time than WT littermates to approach and consume food placed in a novel environment, while GM1-treated YAC128 mice behaved as WT controls (genotype: $F_{1,28}$ = 13.6, $P$ < 0.001; treatment: $F_{1,28}$ = 6.2, $P$ < 0.05; interaction: $F_{1,28}$ = 4.3, $P$ < 0.05; Fig 6B). The anxiolytic effects of GM1 in HD were further confirmed in Q140 mice using the light/dark box test (effect on time in open arm: $F_{2,79}$ = 7.47, $P$ = 0.001; Fig 6C).

Treatment with GM1 also decreased open field defecation in YAC128 ($F_{1,38}$ = 6.980, $P$ < 0.05), Q140 ($F_{2,76}$ = 12.59, $P$ < 0.0001) and R6/2 mice ($F_{1,34}$ = 4.302, $P$ < 0.05) (Fig 6D). Open field defecation is a sympathetic-driven response to anxiogenic stress shared by rodents (Kinn *et al*, 2008; Tache & Brunnhuber, 2008) and humans (Rao *et al*, 1998). Altogether, our data

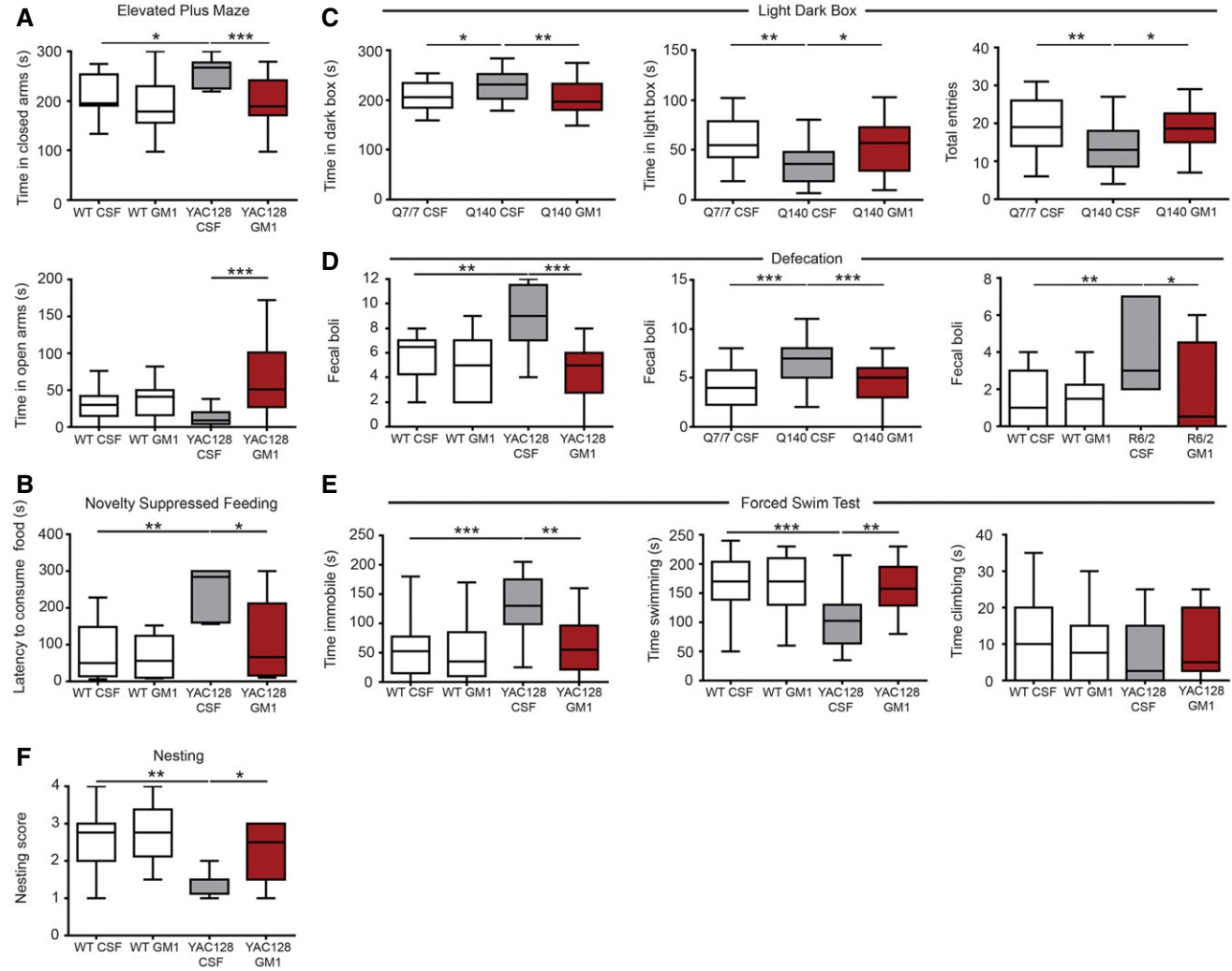

**Figure 6.  GM1 improves anxiety-like and depression-like behaviour in HD mice.**

Mice were treated with cerebro-spinal fluid (CSF, vehicle) or GM1 and tested between day 18–28 (YAC128 mice) or day 25–38 after the beginning of treatment.

A   Time spent by YAC128 mice and wild-type (WT) littermates in the various compartments of an elevated plus maze during a 5-min-long session. N = 15 WT CSF, 14 WT GM1, 13 YAC128 CSF, 19 YAC128 GM1.

B   Novelty-suppressed feeding. Latency to consume sweetened condensed milk in a novel environment is shown. N = 10 WT CSF, 8 WT GM1, 6 YAC128 CSF, 8 YAC128 GM1.

C   Time spent in each compartment of a light–dark box by Q7/7 and Q140. Total number of entries is also shown. N = 28 Q7/7 CSF, 26 Q140 CSF, 29 Q140 GM1.

D   Faecal boli spontaneously excreted in 30-min-long (YAC128 and Q140 mice) or 5-min-long sessions (R6/2) in an open field arena. N = 12 WT CSF, 11 WT GM1, 9 YAC128 CSF, 10 YAC128 GM1; N = 24 Q7/7 CSF, 26 Q140 CSF, 29 Q140 GM1; N = 11 WT CSF, 14 WT GM1, 7 R6/2 CSF, 6 R6/2 GM1.

E   Forced swim test. Time spent swimming, immobile or climbing is shown for 6-month-old YAC128 mice and WT littermates. N = 20 WT CSF, 19 WT GM1, 19 YAC128 CSF, 14 YAC128 GM1.

F   Nest building was scored on a five-point scale (0–4) based on the amount of the nestlet shredded and the height and shape of the nest. N = 12 WT CSF, 12 WT GM1, 8 YAC128 CSF, 10 YAC128 GM1.

Data information: Box-and-whisker plots show median, maximum and minimum values. Two-way (A, B, D, E and F) or one-way ANOVA (C, D) followed by Bonferroni post-test. *$P < 0.05$, **$P < 0.01$, ***$P < 0.001$.

demonstrate that GM1 normalizes anxiety-related behaviours caused by the HD mutation across mouse models with different genetic backgrounds and disease severity.

Depression-like behaviour has been reported in most HD models using the forced swim test (Pla *et al*, 2014). In line with previous reports (Pouladi *et al*, 2009), 6-month-old YAC128 mice spent more time immobile (genotype: $F_{1,67} = 13$, $P < 0.001$) and less time

swimming (genotype: $F_{1,67} = 9.35$, $P < 0.01$) compared to WT littermates, a behaviour that indicates resignation and despair (Cryan *et al*, 2002; Petit-Demouliere *et al*, 2005). Behavioural differences between WT and YAC128 mice were abolished by treatment with GM1 (interaction: $F_{1,67} = 7.3$, $P < 0.01$ for time immobile; interaction: $F_{1,67} = 5.7$, $P < 0.05$ for time swimming; Fig 6E). The therapeutic activity of GM1 was not due to its effects on motor function,

since all mice performed equally well in control tests that measured swimming speed and endurance (Appendix Fig S5A and B). Moreover, acute treatment of YAC128 mice with 10 mg/kg imipramine, a tricyclic antidepressant, dramatically decreased the time YAC128 mice spent immobile and increased time spent swimming, confirming the depression-like nature of YAC128 behaviour in the forced swim test (Appendix Fig S5C). Differently from imipramine, however, GM1 needed to be administered for longer than 10 days to improve depression-like behaviour in YAC128 mice (Appendix Fig S6). Interestingly, GM1 also corrected the behaviour of older WT mice (9 months old; Fig EV2A), which spent more time immobile than younger (6-month-old) mice (41% increase, *t*-test, $P = 0.006$). GM1 also improved the score of YAC128 mice in the nest-building test (genotype: $F_{1,38} = 12.7$, $P < 0.01$; treatment: $F_{1,38} = 5.3$, $P < 0.05$; Fig 6F). This test primarily assesses instinctual species' typical behaviour and general wellness in both male and female rodents (Deacon, 2012), but it can also reveal depression-like behaviour (Nollet *et al*, 2013; Belzung, 2014) and nigrostriatal sensorimotor dysfunction (Sedelis *et al*, 2001; Fleming *et al*, 2004).

As recently reported (Ciamei *et al*, 2015), the performance of Q140 mice in the forced swim test was similar to Q7/7 littermates, but GM1 still decreased time immobile ($F_{2,56} = 6.14$, $P < 0.01$; Fig EV2B). Potential confounding effects related to differences in overall motor activity were excluded with control tests that measured swimming endurance (Fig EV2C).

## GM1 improves cognitive performance in HD mice

Cognitive signs are common in HD and include decreased recognition memory and impaired ability to shift strategy and to plan at earlier disease stages (Paulsen *et al*, 2001b; Snowden *et al*, 2002; Stout *et al*, 2011), as well as deficits in procedural, working and long-term memory with disease progression (Wilson *et al*, 1987; Knopman & Nissen, 1991; Lawrence *et al*, 2000).

In the Crawley's social approach test (Moy *et al*, 2004; Crawley, 2007), all mice showed similar sociability, but YAC128 mice spent more time with a familiar mouse rather than a new mouse (Fig 7A, preference for social novelty), a behaviour that is also observed in AD mouse models (Faizi *et al*, 2012; Cheng *et al*, 2014) and that is generally attributed to altered recognition memory and impaired social discrimination (Moy *et al*, 2004; Faizi *et al*, 2012). Treatment with GM1 restored normal preference for social novelty in YAC128 mice (interaction: $F_{1,53} = 7.275$, $P < 0.01$).

Next, we assessed habituation. This is a complex behaviour that depends on non-associative learning and long-term memory (Leussis & Bolivar, 2006; Bolivar, 2009). Habituation can be measured as the change in exploratory activity that occurs over the time as mice explore a novel open field arena. None of the mice we tested showed intrasession habituation, a measure of non-associative learning (Leussis & Bolivar, 2006; Bolivar, 2009) (indicated by an activity change ratio < 0.5; Fig 7B). Intersession habituation, which depends

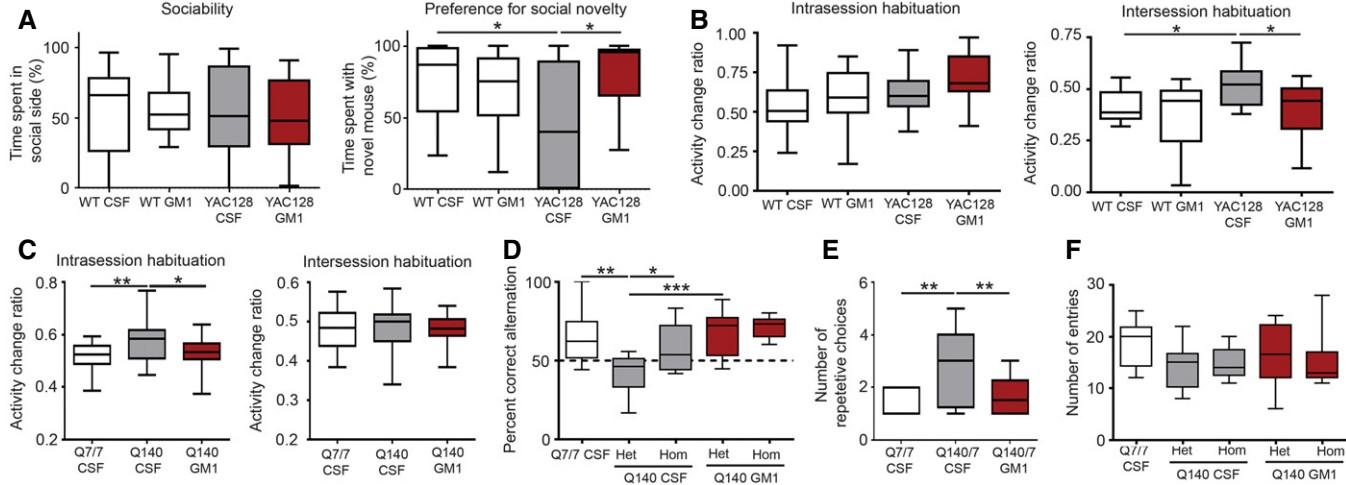

**Figure 7.  GM1 improves the performance of HD mice in social and cognitive tests.**

Mice were treated with artificial cerebro-spinal fluid (CSF, vehicle) or GM1.

A    Three-chamber test. The percent of time mice spent in the chamber containing another mouse in the first part of the test (sociability) is reported. Preference for social novelty in the second part of the test is measured as the time mice spent with a second novel mouse. *N* = 18 WT CSF, 16 WT GM1, 16 YAC128 CSF, 17 YAC128 GM1.

B, C    Mice were placed in an open field apparatus on two consecutive days for 30 min each day. The change in activity between the first and last 5 min of testing on day 1 was used as a measure of intrasession habituation, while the change in total distance travelled between day 1 and day 2 reflected intersession habituation. *N* = 15 WT CSF, 14 WT GM1, 19 YAC128 CSF and 19 YAC128 GM1; *N* = 24 Q7/7 CSF, 24 Q140 CSF, 25 Q140 GM1.

D–F    Y-maze. Mice were allowed to freely explore a Y-maze apparatus for five minutes, during which all arm entries were recorded. Spatial working memory was assessed based on the percent correct alternation mice made (D). A correct alternation was defined as any sequence of three entries where no arm entry was repeated. *N* = 14 WT CSF, 14 Q7/140 CSF, 7 Q140/140 CSF, 14 Q7/140 GM1, 6 Q140/140 GM1. The number of repetitive arm entries (moving back and forth between two arms) is reported in (E) (*N* = 13 WT CSF, 12 Q7/140 CSF, 14 Q7/140 GM1), and the total number of arm entries is reported in (F).

Data information: Box-and-whisker plots show median, maximum and minimum values. Two-way (YAC128 data) or one-way (Q140 data) ANOVA with Bonferroni post-test. *$P < 0.05$, **$P < 0.01$, ***$P < 0.001$.

on memory of the previous session/s (Leussis & Bolivar, 2006; Bolivar, 2009), was evident on the second day of testing in both WT and GM1-treated YAC128 mice (as measured by an activity change ratio below 0.5), but not in vehicle-treated YAC128 animals (genotype: $F_{1,62} = 4.5$, $P < 0.05$; treatment: $F_{1,62} = 6.5$, $P < 0.05$; Fig 7B). Q140 and Q7/7 mice performed poorly in this test and did not show intrasession or intersession habituation (Fig 7C). However, vehicle-treated Q140 mice displayed increased activity at the end of the first 30-min trial (as measured by increased activity change ratio) when compared to other groups ($F_{1,70} = 5.2$, $P < 0.05$). This unusual behaviour, likely due to increased anxiety (Bolivar, 2009), was not present in GM1-treated Q140 mice (Fig 7C).

Next, we assessed the performance of Q7/7 and Q140 mice in the Y-maze (Kokkinidis & Anisman, 1976; Paul et al, 2009; Fig 7D–F). Only data for male mice are shown, as all females scored similarly in this test, regardless of genotype and treatment. Because in this test we observed a difference in behaviour between heterozygous and homozygous Q140 mice, data for the two groups are shown separately. As expected, Q7/7 mice scored above chance level (correct alternation rate > 50%; chi-square test, $P < 0.05$). The average alternation rate for vehicle-treated heterozygous Q7/140 mice was significantly worse ($F_{4,50} = 8.1$, $P < 0.01$; Fig 7D), and these mice made significantly more repetitive choices than Q7/7 mice prior to performing a correct alternation ($F_{2,32} = 6.373$, $P < 0.01$; Fig 7E). This behaviour—which can also be observed in models of obsessive-compulsive disorders (Yadin et al, 1991; Ulloa et al, 2004) or following impairment of serotonergic pathways (Geyer et al, 1976)—is indicative of perseveration, or cognitive inflexibility (Kokkinidis & Anisman, 1977; Kokkinidis, 1987), and reflects a common executive dysfunction in HD patients (Lawrence et al, 1996; Paulsen, 2011). Treatment with GM1 restored normal behaviour (Fig 7D and E). The total number of arm entries was similar across experimental groups, indicating similar exploratory and locomotor activity in this test (Fig 7F). Surprisingly, vehicle-treated homozygous Q140/140 mice performed as well as Q7/7 in this test.

## GM1 treatment induces neurochemical changes in the brain of HD mice

Changes in levels of glutamate, γ-aminobutyric acid (GABA; Pearson & Reynolds, 1994; Rosas et al, 2008; Estrada-Sanchez & Rebec, 2013; Padowski et al, 2014) and monoamines (Yohrling et al, 2002; Dang et al, 2012; Du et al, 2013; Pla et al, 2014; Vinther-Jensen et al, 2015) have been linked to psychiatric and cognitive dysfunctions in HD and can affect performance in the behavioural tests performed in this study (Detke et al, 1995; Cryan et al, 2005; Leussis & Bolivar, 2006). Therefore, we sought to determine whether the beneficial effects of GM1 on HD mouse behaviour were accompanied by neurochemical changes. Neurochemical analysis was performed in male mice only, to avoid potential confounding effects of oestrous cycle on the dopaminergic system (Thompson & Moss, 1997; Yu & Liao, 2000).

Levels of glutamate were lower in cortical tissue from YAC128 mice compared to WT (genotype: $F_{1,26} = 6.7$, $P < 0.05$; Fig 8), likely reflecting early cortical pathology in mice and in HD patients (Rosas et al, 2002, 2008, 2011; Estrada-Sanchez & Rebec, 2013). Our study also revealed a small but significant decrease in GABA levels in the cortex of YAC128 mice compared to WT, in line with earlier studies

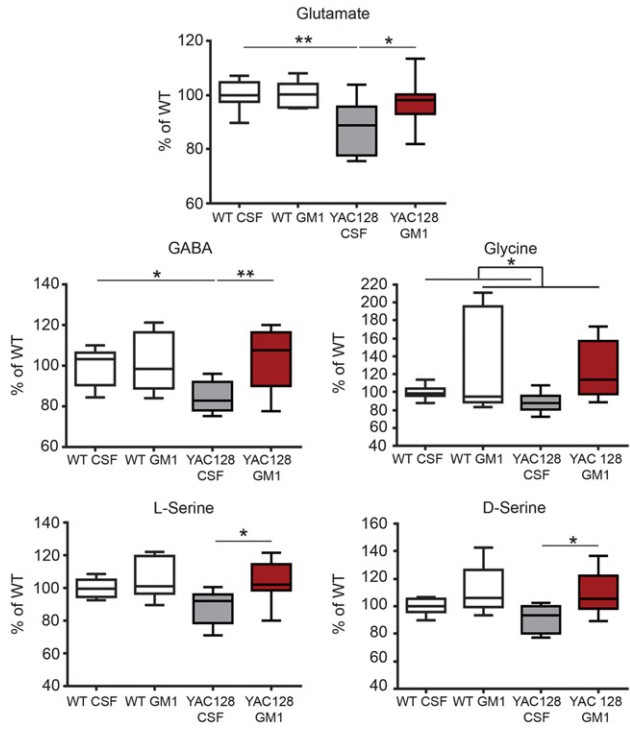

**Figure 8.  GM1 restores wild-type levels of neuroactive amino acid levels in the cortex of YAC128 mice.**

Amino acid levels were measured in the cortex of YAC128 mice and are expressed as percentage of WT control (CSF-treated). $N$ = 8 WT CSF, 7 WT GM1, 8 YAC128 CSF, and 8 YAC128 GM1, with the exception of Glu and GABA in the WT GM1 group ($N$ = 6) and GABA in the YAC128 CSF group ($N$ = 7).

Data information: Box-and-whisker plots show median, maximum and minimum values. Two-way ANOVA with Bonferroni post-test. *$P < 0.05$, **$P < 0.01$.

in HD patients (Pearson & Reynolds, 1994), and with more recent reports of reduced GABAergic inhibitory input from cortical interneurons in HD models (Gu et al, 2005; Spampanato et al, 2008; Cummings et al, 2009). Administration of GM1 restored control levels of cortical glutamate and GABA ($F_{1,27} = 5.2$, $P < 0.05$), and it also increased levels of glycine ($F_{1,27} = 7.0$, $P < 0.05$), D-serine ($F_{1,27} = 10.2$, $P < 0.01$) and L-serine ($F_{1,27} = 6.8$, $P < 0.05$; Fig 8), suggesting widespread disease-modifying effect of GM1 in YAC128. In the striatum, amino acid levels were similar across groups (data not shown).

Psychiatric and cognitive problems—including cognitive inflexibility—have been linked to changes in monoaminergic systems in both HD and non-HD populations (Castro et al, 1998; Yohrling et al, 2002; Du et al, 2013; Dang et al, 2012). As summarized in Table 1, treatment with GM1 resulted in region-dependent changes in monoamines and/or their catabolites. In the striatum—where dopamine (DA) and metabolites resulting from DA turnover, 3,4-dihydroxyphenylacetic acid (DOPAC) and homovanillic acid (HVA), were decreased compared to WT—there was a significant effect of GM1 treatment on DA (increased by GM1) and DOPAC levels (decreased by GM1) in both WT and YAC128 mice. At the same time, GM1 decreased DOPAC/DA and DOPAC + HVA/DA ratios, suggesting an effect of this ganglioside on DA turnover. In

**Table 1. Effect of GM1 treatment on biogenic amines.**

| | WT | | YAC128 | | Two-way ANOVA (Bonferroni post-test) |
|---|---|---|---|---|---|
| | CSF | GM1 | CSF | GM1 | |
| **Striatum** | | | | | |
| DA | $100 \pm 8.7$ | $105.2 \pm 6.6$ | $88.2 \pm 12.0^v$ | $99.2 \pm 11.3$ | Effect of genotype: $F(1, 27) = 6.16$, $P < 0.05$; Effect of treatment: $F(1, 27) = 5.092$, $P < 0.05$<br>v: HD CSF ≠ WT GM1 |
| DOPAC | $100 \pm 13.7$ | $81.5 \pm 9.8^§$ | $78.0 \pm 12.7^{**}$ | $66.0 \pm 13.9^{\#\#\#}$ | Effect of genotype $F(1, 27) = 16.8$, $P < 0.001$; Effect of treatment $F(1, 27) = 11.11$, $P < 0.01$<br>**: HD CSF ≠ WT CSF; §: WT GM1 ≠ WT CSF;<br>###: HD GM1 ≠ WT CSF |
| HVA | $100 \pm 11.7$ | $97.9 \pm 9.6$ | $81.3 \pm 6.7^{**,v}$ | $84.0 \pm 12.6^{\#}$ | Effect of genotype $F(1, 27) = 18.92$, $P < 0.001$<br>**: HD CSF ≠ WT CSF; v: HD CSF ≠ WT GM1; #: HD GM1 ≠ WT CSF |
| DOPAC/DA | $100 \pm 18$ | $77.0 \pm 12.4^§$ | $88.0 \pm 13.6$ | $66.3 \pm 13^{\wedge,\#\#\#}$ | Effect of genotype $F(1, 27) = 4.704$, $P < 0.05$; Effect of treatment $F(1, 27) = 18.22$, $P < 0.001$<br>§: WT GM1 ≠ WT CSF; ^: HD GM1 ≠ HD CSF;<br>###: HD GM1 ≠ WT CSF |
| HVA/DA | $100 \pm 7.8$ | $93.2 \pm 9.8$ | $92.8 \pm 7.8$ | $85.3 \pm 15^{\#}$ | #: HD GM1 ≠ WT CSF |
| 5HT | $100 \pm 14.8$ | $97.5 \pm 11.9$ | $92.9 \pm 22.1$ | $93.3 \pm 11.0$ | N.S. |
| 5HIAA | $100 \pm 11.3$ | $90.5 \pm 11.3$ | $85.1 \pm 8.3^{*}$ | $76.9 \pm 9.3^{\#\#\#}$ | Effect of genotype $F(1, 27) = 15.35$, $P < 0.001$; Effect of treatment: $F(1, 27) = 5.961$, $P < 0.05$<br>*: HD CSF ≠ WT CSF; ###: HD GM1 ≠ WT CSF |
| 5HIAA/5HT | $100 \pm 9.5$ | $92.7 \pm 10.2$ | $95.8 \pm 26.8$ | $83.1 \pm 17.2$ | N.S. |
| NA | $100 \pm 39.6$ | $133.9 \pm 64.4$ | $137.3 \pm 67.9$ | $120.7 \pm 42.1$ | N.S. |
| **Cortex** | | | | | |
| DA | $100 \pm 32.7$ | $113.6 \pm 19.0$ | $106.4 \pm 27.6$ | $97.5 \pm 14.1$ | N.S. |
| DOPAC | $100 \pm 26.7$ | $123.1 \pm 31.0$ | $142.1 \pm 31.7^{**}$ | $92.1 \pm 16.0^{\wedge\wedge}$ | Interaction $F(1, 27) = 14.26$, $P < 0.001$<br>**: HD CSF ≠ WT CSF; ^^: HD GM1 ≠ HD CSF |
| HVA | $100 \pm 9.9$ | $118.3 \pm 17.9^§$ | $105.6 \pm 17.6$ | $97.2 \pm 11.9^{\infty\infty}$ | Interaction: $F(1, 27) = 6.403$, $P < 0.05$<br>§: WT GM1 ≠ WT CSF; ∞: HD GM1 ≠ WT GM1 |
| DOPAC/DA | $100 \pm 33.3$ | $100.0 \pm 19.4$ | $133.3 \pm 57.6$ | $88.0 \pm 5.9^{\wedge}$ | ^: HD GMI ≠ HD CSF |
| HVA/DA | $100 \pm 19.5$ | $96.5 \pm 16.6$ | $95.7 \pm 25.9$ | $94.7 \pm 15.1$ | N.S. |
| 5HT | $100 \pm 14.0$ | $101.6 \pm 12.8$ | $84.5 \pm 19.2$ | $105.4 \pm 7.5^{\wedge}$ | Effect of treatment: $F(1, 26) = 4.623$, $P < 0.01$<br>^: HD GM1 ≠ HD CSF |
| 5HIAA | $100 \pm 9.7$ | $102.7 \pm 19.1$ | $104.5 \pm 17.0$ | $83.6 \pm 15.8^{\wedge}$ | Interaction $F(1, 27) = 4.369$, $P < 0.05$<br>^: HD GM1 vs. HD CSF |
| 5HIAA/5HT | $100 \pm 23.0$ | $98.4 \pm 17.5$ | $129.0 \pm 52.2$ | $80.4 \pm 16.7^{\wedge}$ | Effect of treatment: $F(1, 27) = 4.911$, $P < 0.05$; Interaction $F(1, 27) = 4.324$, $P < 0.05$<br>^: HD GM1 ≠ HD CSF |
| NA | $100 \pm 19.9$ | $94.3 \pm 12.9$ | $95.3 \pm 11.5$ | $98.0 \pm 14.8$ | N.S. |

N.S. = Not Significant.
Biogenic amine levels were expressed as percentage of WT control (CSF). Values are means ± STDEV. Symbols used to indicate statistically significant differences between groups ($P < 0.05$) were repeated twice or three times to indicate different $P$-values ($P < 0.01$ and $P < 0.001$, respectively). For all amines measured unless otherwise indicated: $N = 8$ WT CSF, 7 WT GM1, 8 HD CSF, 8 HD GM1. $N = 7$ for cortical 5-HT measured in the HD GM1 group. $N = 6$ for striatal NA measured in the WT GM1 group.

the cortex, levels of DA were similar across groups (Table 1). DOPAC levels were significantly elevated in vehicle-treated YAC128 mice compared to WT, and normalized by GM1 treatment. Therefore, GM1 treatment appeared to have an overall modulatory action on the dopaminergic system and dopamine turnover, with the direction of the effects likely being dependent on overall tissue dopaminergic tone.

Serotonin (5HT) and its metabolites were also affected by GM1 administration. GM1 increased cortical, but not striatal,

5HT levels in YAC128 mice, perhaps depending on local serotonergic tone. Concomitantly, GM1 decreased 5-hydroxyindoleacetic acid (5HIAA), a product of serotonin catabolism, both in the striatum and in the cortex of YAC128 mice. Overall, these changes suggest slower or decreased 5HT turnover (San-Martin-Clark *et al*, 1993) as a result of GM1 administration and are in line with the antidepressant effects of GM1 in HD models. Levels of noradrenaline were normal in YAC128 mice and not affected by GM1 (Table 1).

# Discussion

Our studies provide evidence of widespread therapeutic and disease-modifying effects of GM1 across different HD mouse models, independently from mouse strain and genetic make-up.

In R6/2 mice, GM1 administration decreased striatal atrophy and neuronal loss, hallmarks of HD that correlate with disease severity and progression in HD patients (Tabrizi et al, 2011, 2012). The number of striatal neurons in R6/2 mice that received GM1 was restored to WT levels, suggesting that the relatively early start of treatment (6 weeks of age), when neurodegeneration is still modest in this mouse model (Dodds et al, 2014), prevented neuronal loss during the period of treatment. Whether neurogenesis might have also contributed to the maintenance of striatal neuron number remains to be determined.

GM1 had trophic effects on corpus callosum and cortico-striatal white matter tracts. Improved brain connectivity through these structures would predict cognitive improvement (Crawford et al, 2013; Novak et al, 2014; Matsui et al, 2015), which was in fact observed upon administration of GM1 in YAC128 and Q140 mice. GM1 is found in high concentrations in white matter in the mouse brain (Vajn et al, 2013), and gangliosides are required for the maintenance of myelinated axons (Yang et al, 1996; Sheikh et al, 1999; Pan et al, 2005). Whether in our study GM1 exerted a trophic/protective action on oligodendrocytes and myelin directly, or indirectly, through improved neuronal health, remains to be investigated.

One of the most striking changes observed in R6/2 mice upon administration of GM1 was the reduction of ferritin accumulation in the brain. Ferritin levels are an indirect measure of iron accumulation, which correlates, in HD patients, with CAG repeat length and with the degree of cortical and basal ganglia atrophy (Muller & Leavitt, 2014; Sanchez-Castaneda et al, 2015). Furthermore, in R6/2 mice, high levels of ferritin and ferric iron accumulate within dystrophic microglia and elicit pro-inflammatory microglia activation (Simmons et al, 2007). Thus, the ability of GM1 treatment to reduce ferritin levels in R6/2 mice might lead to decreased oxidative stress and damage, as well as to an improvement of microglia function.

The role of microglia in HD pathogenesis is still unclear. Reactive microglia are found in all grades of HD pathology, including pre-symptomatic patients, in correlation with increased levels of pro-inflammatory cytokines (Dalrymple et al, 2007; Bjorkqvist et al, 2008; Silvestroni et al, 2009; Politis et al, 2011). However, the relative contribution of microglial cell autonomous dysfunction (Bjorkqvist et al, 2008; Crotti et al, 2014) compared to microglial changes that develop secondary to the disease process in HD is still unclear. Conflicting findings have been reported for R6/2 mice, with some studies describing microglia activation (Stack et al, 2006; Giampa et al, 2010) and another reporting decreased microglia density (Laurine et al, 2003). In our study, and in spite of ferritin accumulation, we did not detect inflammatory markers nor an increase in the number of microglia cells in R6/2 mice; in cortical tissue, we actually observed decreased expression of Iba1, a microglia marker and calcium-binding protein important for phagocytosis and microglia motility (Kanazawa et al, 2002). This observation is in line with, and could contribute to explain, reports of abnormal microglia motility in HD (Kwan et al, 2012). Furthermore, it suggests the presence of a previously unrecognized aberrant "molecular" signature (or activity state) of R6/2 microglia that develops

independently from pro-inflammatory changes and that we speculate could potentially affect protective housekeeping functions of microglia. This conclusion is also supported by our data showing decreased expression of Spi-1—the gene that encodes the master regulator of microglia differentiation and function (Heinz et al, 2010; Kierdorf et al, 2013)—in R6/2 mice. The apparent discrepancy between our data and previous reports that showed increased expression of Spi-1 in R6/2 and other HD models (Crotti et al, 2014) might reflect differences in the overall health status of the mouse colonies used in different laboratories, and/or differences in the amount of environmental enrichment provided, since both these factors are known to affect microglia phenotype (Bilbo et al, 2007; Hoogland et al, 2015; Rodríguez et al, 2015). Importantly, GM1 treatment restored Spi-1 expression levels to normal and increased Iba-1 levels in the striatum of R6/2 mice.

We also observed an effect of GM1 on the expression of GFAP, an intermediate filament protein with various functions in astrocytes (Bartzokis et al, 2007a) that has been proposed to have a role in the regulation of blood–brain barrier and myelin integrity as well as in the pathophysiology of depression (Middeldorp & Hol, 2011). We found that in spite of a slight increase in GFAP-immunoreactive area in the cortex of CSF-treated R6/2 mice, GFAP protein expression was lower than normal in cortical lysates from these mice. Similar findings were previously described in a rat model of HD (Cong et al, 2012). Although the overall implications of these findings remain to be elucidated, studies in a human astrocytoma cell line indicate that reduced GFAP expression affects astrocyte maturation and their ability to respond to neurons (Weinstein et al, 1991). The fact that GM1 treatment restored GFAP protein expression to wild-type levels suggests a beneficial effect on astrocyte function. In light of the emerging roles of non-neuronal cell populations in HD pathogenesis (Lobsiger & Cleveland, 2007; Ehrlich, 2012), we speculate that the effects exerted by GM1 on HD astrocytes and microglia may contribute to the disease-modifying properties of this ganglioside in HD models.

Striatal dysfunction precedes striatal atrophy in HD patients and animal models, and is associated with the disruption of signalling pathways that converge on DARPP32 (Bibb et al, 2000; Svenningsson et al, 2004). Expression and phosphorylation (Thr34) of DARPP32 were restored to wild-type levels after treatment with GM1 in heterozygous Q7/140 mice but not in homozygous Q140/140. The inability of GM1 to increase DARPP32 levels in the latter is intriguing and might point at the requirement for expression of some wild-type HTT to mediate the effects of GM1 on DARPP32. Other beneficial effects of GM1 on molecular and behavioural HD phenotypes occurred in both Q7/140 and Q140/140 mice, suggesting that expression of wild-type HTT was not required in all cases. Interestingly, we found that in Q140 mice, restoration of normal behaviour required a longer treatment with GM1 (28 days or longer) compared to R6/2 and YAC128 mice (Di Pardo et al, 2012), where significant benefits were already measurable after 14 days of treatment. Whether this was due to the different genetic background of these mice, or reflects the involvement of specific HTT functions (and therefore the requirement for wild-type HTT expression) in mediating at least some of GM1 benefits, remains to be investigated.

A crucial finding in our studies was that GM1 treatment affects mHTT itself. We previously showed that GM1 administration modifies levels of HTT phosphorylation at Ser13 and Ser16 (Di Pardo

*et al*, 2012), a post-translation modification associated with a reduction in mHTT aggregation (Gu *et al*, 2009) and toxicity (Gu *et al*, 2009; Thompson *et al*, 2009; Atwal *et al*, 2011). In line with these reports, our study provides evidence that administration of GM1 decreases levels of soluble and aggregated (SDS-insoluble) mHTT, without affecting wild-type HTT levels. Since the expression of HTT mRNA was not altered by GM1, we speculate that the observed reduction of mHTT levels was likely due to increased protein clearance. The underlying mechanism remains to be investigated. GM1 might activate autophagy, as was shown in other disease models (Hwang *et al*, 2010; Garofalo *et al*, 2016; Dai *et al*, 2017), thus potentially facilitating autophagic degradation of mutant HTT. Future experiments will test this hypothesis.

The disease-modifying effects of GM1 were mirrored by a profound improvement of motor and non-motor dysfunctions in HD mice. In Q140 mice, motor skills were restored by GM1 to wild-type levels, including gait abnormalities, which are often resistant to treatments that improve other motor symptoms (Ferrara *et al*, 2012; Kegelmeyer *et al*, 2014). In R6/2 mice, motor improvement was not as marked as in Q140 mice, but comparable to or better than that reported in another study where expression of mHTT had been targeted (decreased) in the same animal model (Cheng *et al*, 2015). Modest motor improvement in R6/2 mice is not surprising considering the much more widespread neuropathology and disease severity that characterize this model. Nevertheless, the behavioural effects of GM1 treatment in R6/2 mice were consistent with a disease-modifying action of this ganglioside. While a rapidly progressing motor dysfunction characterized CSF-treated R6/2 mice, those that received GM1 maintained stable performance throughout the study. In line with these findings, GM1 also prevented weight loss in R6/2 mice.

In addition to motor symptoms, GM1 corrected a large spectrum of emotionality-related and cognitive phenotypes—along with underlying neurochemical defects—that are particularly difficult to treat in HD and that are often overlooked in pre-clinical studies. Restorative effects of GM1 administration on anxiety-like behaviour in HD mice were measured with a battery of tests that make use of anxiogenic paradigms that are based on distinct—although overlapping—neurobiological substrates (Cryan & Holmes, 2005; Ramos, 2008). GM1 administration was also able to decrease depression-like behaviour in HD mice. Interestingly, GM1 decreased a depression-like behaviour that developed in WT mice with ageing (Turner *et al*, 2012). These data suggest that GM1 targets pathways that are affected in depressive behaviour and that are shared between HD and ageing WT mice. GM1 also improved the performance of HD mice in the novelty-suppressed feeding test and in nest building. Although not specific for depression, both these tests are affected by and can reveal depression-like behaviours in rodents (Santarelli *et al*, 2003; David *et al*, 2009; Samuels & Hen, 2011; Nollet *et al*, 2013; Belzung, 2014). Therefore, the improved performance of HD mice in these tests after treatment with GM1 might be due to a decrease of both anxiety and depression.

In the forced swim test, GM1 treatment increased the time mice spent swimming, but not the time spent climbing. This is reminiscent of the effects of selective serotonin reuptake inhibitors (SSRI) and is in contrast to the effects of antidepressants that elevate noradrenaline levels (which increase time spent climbing; Detke *et al*, 1995; Cryan *et al*, 2005). In line with these observations, GM1 increased cortical levels of serotonin but not adrenaline in YAC128

mice (Table 1). Concomitantly, GM1 decreased 5HIAA levels, suggesting decreased serotonin turnover (San-Martin-Clark *et al*, 1993). In addition to its effects on serotonin turnover and levels, we speculate that treatment with GM1 could potentially improve overall serotonergic transmission, since this and other gangliosides were shown to bind serotonin and facilitate its interaction with serotonin receptors (Yandrasitz *et al*, 1980; Matinyan *et al*, 1989; Fantini & Barrantes, 2009), and to increase functional coupling of serotonin receptors with adenylate cyclase (Berry-Kravis & Dawson, 1985). Altogether, these data suggest that GM1 has a modulatory effect on the serotonergic system that could at least in part explain its antidepressant effects in animal models.

Administration of GM1 improved cognitive behaviour in both YAC128 and Q140 mice, as shown by improved performance in Crawley's test, Y-maze and open field habituation test. Administration of GM1 also improved intersession habituation in HD mice, a complex learning behaviour that is also compromised in AD models (Muller *et al*, 1994) and by drugs that affect memory (Platel & Porsolt, 1982; Hess *et al*, 1986). Serotonergic, cholinergic and glutamatergic activities play a major role in the habituation process (Leussis & Bolivar, 2006). Although we did not measure acetylcholine levels, in our study, GM1 increased serotonin, glutamate, D-serine (an NMDA receptor co-agonist) and its precursor L-serine, a finding that contributes to explain the beneficial effects of GM1 on habituation in HD mice.

Psychiatric and cognitive problems, including cognitive inflexibility, have been linked to changes in monoaminergic systems in both HD and non-HD populations (Yohrling *et al*, 2002; Dang *et al*, 2012; Du *et al*, 2013). The exact contribution of the DA system to HD pathology remains unclear (Chen *et al*, 2013b; Schwab *et al*, 2015). Although DA release is decreased in HD animal models (Hickey *et al*, 2002; Johnson *et al*, 2006; Callahan & Abercrombie, 2011), in HD patients, the DA system undergoes more complex dynamic and biphasic changes (Spokes, 1980; Kish *et al*, 1987; Garrett & Soares-da-Silva, 1992). In this study, we found decreased DA levels in the striatum of YAC128 mice compared to WT. We also observed region-specific differences in the levels of DA metabolites between YAC128 and WT littermates, with DOPAC (and HVA) being decreased in the YAC128 striatum, but increased in the YAC128 cortex. These findings might reflect changes in MAOs expression and/or activity, as reported in other studies (Richards *et al*, 2011; Ooi *et al*, 2014; Laprairie *et al*, 2015), while the direction of these changes (whether up or down) might depend on regional differences in dopaminergic tone. Of note, polymorphisms in the genes that code for monoamine oxidase A (*MAOA*) and catechol-O-methyl-transferase (*COMT*) have been recently shown to act as modifiers of cognitive and psychiatric symptoms in a Danish HD population (Vinther-Jensen *et al*, 2015), thus strengthening the link between MAOA activity, dopaminergic tone and cognitive and psychiatric manifestations of HD.

GM1 treatment had variable effects on DA turnover, depending on brain region and mouse genotype. GM1 decreased DOPAC and DOPAC/DA ratio to WT levels in the cortex of YAC128 mice, but further decreased DOPAC and DOPAC/DA ratio below WT levels in the striatum. GM1 also affected DOPAC levels (and DOPAC/DA ratio) in WT animals. We speculate that changes induced by GM1 might reflect a modulatory and stabilizing—rather than inhibitory—action of GM1 on DA turnover. A modulatory role of GM1 was also

suggested in previous studies, where administration of GM1 was shown to increase dopaminergic transmission in aged rats, but not in young animals (Goettl *et al*, 2003).

To the best of our knowledge, the widespread therapeutic and disease-modifying effects of GM1 observed in our studies are only comparable to (and even exceed in certain cases) those exerted by antisense oligonucleotide therapies to decrease HTT production (Kordasiewicz *et al*, 2012). Such profound effects are likely the result of GM1 effects on HTT phosphorylation and overall levels, as discussed above. These crucial disease-specific effects of GM1 are likely to be accompanied by additional general neuroprotective activities known to be exerted by this ganglioside (Mocchetti, 2005; Posse de Chaves & Sipione, 2010; Ledeen & Wu, 2015). The therapeutic effects of GM1 shown in this study warrant clinical investigations in HD patients. Past clinical trials for the treatment of other conditions have shown that GM1 administration in patients is relatively safe (Alter, 1998; Chinnock & Roberts, 2005; Schneider *et al*, 2010, 2013). In spite of poor permeability of the blood–brain barrier to GM1 (Ghidoni *et al*, 1986; Saulino & Schengrund, 1994), sustained benefits of GM1 were observed in a small randomized double-blind placebo-controlled trial in Parkinson's disease patients even when this ganglioside was administered by subcutaneous injections (Schneider *et al*, 2010, 2013, 2015). Potential drug delivery challenges in HD patients could be circumvented through intrathecal (or intraventricular) drug administration, as in recent clinical trials with antisense oligonucleotides, or with strategies and formulations to improve drug delivery to the brain. Co-administration of GM1 with HTT-lowering agents could also be envisioned, to increase therapeutic benefits and potentially extend "HTT holiday" periods.

# Materials and Methods

### Study design

Controlled animal experiments were performed in YAC128, Q140 and R6/2 mice.

Detailed information on mouse strains is provided in the Appendix—Materials and Methods. Throughout the paper, the abbreviation Q140 is used to refer to both Q7/140 heterozygous and Q140/140 homozygous mice, while wild-type littermates are referred to as Q7/7. All mice were maintained in our animal facility at the University of Alberta on a 14-/10-h light–dark cycle in a temperature- and humidity-controlled room. All procedures on animals were approved by the University of Alberta's Animal Care and Use Committee and were in accordance with the guidelines of the Canadian Council on Animal Care.

Chronic intraventricular administration of semi-synthetic GM1 (provided by Seneb BioSciences Inc., Holliston, MA) or vehicle (artificial cerebro-spinal fluid, CSF, Harvard Apparatus) was performed as previously published (Di Pardo *et al*, 2012), with minor modifications (detailed in the Appendix—Materials and Methods). Treatment duration was 28 days for YAC128 and R6/2 mice and 42 days for Q140 mice. Mouse age at the start of treatment was: 6 (cohort 1) or 8 weeks (cohort 2) for R6/2 mice, $6.4 \pm 0.86$ months for Q140, $6.4 \pm 0.35$ (cohort 1) and $9.2 \pm 0.37$ (cohort 2) months for YAC128 mice (see also Appendix Fig S1 for further experimental details). All mice were already symptomatic (as assessed by motor tests) at the

beginning of treatment. R6/2 mice and their WT littermates were all males. Both male and female Q140 mice were used. In most tests, no significant sex-specific differences in behaviour were observed; therefore, data from males and females were combined. The only exceptions were as follows: the rotarod test, where male mice showed no impairment compared to Q7/7 littermates and were therefore excluded from the analysis, and the Y-maze and the nest-building tests, where female Q140 mice show no deficit and were excluded from the analysis. Data from homozygous and heterozygous Q140 mice were combined, unless otherwise indicated, as these mice performed similarly in most tests (see also Appendix Table S1 for statistical analysis). All mice were housed individually throughout the period of treatment to avoid accidental displacement of the infusion kit and/or wound infection.

### Histology and volumetric analysis

Mice were euthanized by cervical dislocation, and brains were immediately removed and flash-frozen in isopentane (2-methylbutane, Acros Organics) pre-cooled at $-80°C$ and kept on dry ice. Brains were then stored at $-80°C$ until cryosectioning. Twenty micrometre-thick serial coronal sections were obtained using a cryostat and thaw-mounted onto charged slides (Superfrost Plus, Fisher). Slides were left to dry at RT overnight, prior to storage at $-20°C$. All sections were post-fixed in formalin (10% buffered formalin phosphate, Fisher) for 5 min at RT prior to use. Volumetric analyses were performed from serial photomicrographs taken with an Olympus BX60 microscope coupled to a Cool Snap Image Pro colour camera, using the Cavalieri principle (Henery & Mayhew, 1989). Details are provided in the Appendix—Materials and Methods.

### Immunohistochemistry and immunofluorescence staining

For immunohistochemistry, sections were dehydrated in serial alcohol dilutions, cleared with xylene and then rehydrated. Endogenous peroxidase activity was quenched in 1% hydrogen peroxide in 50% methanol for 10 min. Sections were then blocked in 0.3% Triton X-100 (for NeuN staining) or 0.2% Triton X-100 (for ferritin staining) in Universal Blocker (DakoCytomation) for 20 min (NeuN) or 1 h (ferritin) at RT, before incubating with the primary antibody overnight at 4°C. Incubation with biotinylated secondary antibodies was for 30 min at RT where the primary antibody was not conjugated to biotin. The signal was amplified by incubation with the avidin biotin complex (ABC) system for 30 min at RT and then detected with diaminobenzidine (DAB) (Sigma). Slides were mounted with Permount (Fisher Scientific. Waltham, MA). For immunofluorescence staining, brain sections were blocked with 0.2% Triton X-100 (Fluka) in Universal Blocker (DakoCytomation) for 1 h at RT and then incubated in primary antibodies overnight in a humidity chamber at 4°C. After washing with phosphate-buffered saline (PBS), sections were incubated with secondary antibodies for 30 min at RT, washed and mounted in ProLong Gold mounting media (Life Technologies).

### Eriochrome staining

To identify white matter structures, sections were stained with Eriochrome Cyanine R as previously described (Kiernan, 1984) with

minor modifications as detailed in the Appendix—Materials and Methods.

## Tissue lysate preparation and immunoblotting

Brains were dissected immediately after mouse euthanasia and flash-frozen in liquid $N_2$. For immunoblotting, samples were immediately homogenized in ice-cold lysis buffer (20 mM Tris, pH 7.4, 1% Igepal, 1 mM EDTA, 1 mM EGTA, 50 μM MG132, 1× Roche complete protease inhibitor cocktail and 1× Roche PhosStop phosphatase inhibitor cocktail), using a motor-driven Potter-Elvehjem homogenizer. Samples were sonicated twice for 10 s each using a Sonic Dismembrator Model 100 and then centrifuged at 20,000 *g* for 10 min at 4°C. Protein concentration in the supernatants was measured using the bicinchoninic acid (BCA) assay (Thermo). Thirty micrograms of proteins was separated by electrophoresis on a 4–20% SDS–polyacrylamide gel and transferred onto Immobilon-FL polyvinylidene fluoride (PVDF) membrane (Millipore). For HTT immunoblotting, proteins were separated in 4–12% SDS–polyacrylamide gels and transferred overnight onto an Immobilon-FL PVDF membrane in transfer buffer containing 0.01% sodium dodecyl sulphate (SDS) and 16% methanol. All membranes were blocked with 5% bovine serum albumin (BSA) in Tris-buffered saline containing 0.1% Tween-20 (TBS-T) and incubated overnight at 4°C with primary antibodies, followed by the appropriate IRDye secondary antibody (1:40,000, LI-COR Biotechnology) for 45 min at RT. Infrared signal was acquired and quantified using the Odyssey Imaging System. Swift™ total protein stain (G Biosciences) was performed according to manufacturer's instructions as the loading control.

## Filter retardation assay

Filter retardation assay was performed as described in (Wanker *et al*, 1999), with minor modifications. Briefly, 30 μg of protein lysates were diluted in PBS, denatured and reduced by adding 2% SDS and 100 mM DTT to the lysis buffer, followed by heating at 98°C for 10 min. Samples were filtered through a cellulose acetate membrane (0.2 μm pore size, Sterlitech) in a Bio-Dot microfiltration unit (Bio-Rad). Wells were washed twice with 200 μl PBS. After drying for 30 min, membranes were washed twice with 2% SDS in PBS and then blocked with 5% BSA in TBS-T followed by incubation with anti-HTT antibodies (antibodies used and dilutions are detailed in the Supplementary Material and Methods), and appropriate IRDye secondary antibodies (LI-COR Biotechnology) were used at 1:20,000 for 1 h at RT. Infrared signal was acquired and quantified using the Odyssey Imaging System.

## Quantitative polymerase chain reaction (qPCR)

RNA extraction was performed from brain tissue preserved in RNAlater Solution (Ambion), using a RNA extraction kit (Qiagen) according to manufacturer's instruction. cDNA was produced using SuperScript II Reverse Transcriptase (Thermo) and amplified with Power Up SYBR Green PCR Master Mix (Invitrogen). Quantitative PCR was carried out in a StepOnePlus instrument using the following primers: *Htt* Fw: 5′-GGGAAAGAGCTTGAGACACAG, *Htt* Rev: 5′-AAGGATGAACATCTCCAGCAC; and *Spi-1* Fw: 5′-ATGGAGAAGCT GATGGCTTG, *Spi-1* Rev: 5′-AGGTCCAGCAGGAACTGGTA. Mouse

*Atp5B*, *Cyclophilin A* and *eIF4a2* were used as reference genes and their geometric mean was used to calculate a normalization index (Pfaffl *et al*, 2004). Target gene expression was calculated using the relative standard curve method.

## Analysis of biogenic amines and amino acids

At the end of treatment, mice were euthanized by cervical dislocation. Cortical and striatal tissues from the left brain hemisphere (contralateral to the site of cannulation) were collected, flash-frozen in liquid nitrogen and stored at −80°C prior to neurochemical analysis. Amino acids were analysed by HPLC according to (Grant *et al*, 2006) with minor modifications. Tissues were homogenized in 5 volumes of MeOH, let sit on ice for 10 min and then centrifuged at 10,000 *g* for 4 min. Supernatants were diluted up to 30- or 60-fold in Milli-Q system-filtrated water. Aliquots of the diluted material were derivatized with o-phthaldialdehyde (OPA, Sigma-Aldrich) and N-isobutyryl-L-cysteine (IBC, Novachem) for HPLC analysis. Fluorescence detector was set at an excitation wavelength of 344 nm and emission at 433 nm. Biogenic amines and their metabolites were analysed according to Parent *et al* (2001). Briefly, $1/10^{th}$ the volume of ice-cold 1 N $HClO_4$ containing 500 μM ascorbic acid and EDTA (100 mg%) was added to tissue aliquots in 5 volumes of water. Samples were then vortexed and centrifuged at 10,000 *g* for 4 min. Supernatants were used for HPLC analysis. Calibration curves were constructed for each HPLC run. Electrochemical detection was performed with an applied potential of 0.65V.

## Analysis of behaviour

Behavioural testing was conducted in the light phase of the cycle between 08:00 h and 18:00 h. All tests were performed by experimenters who were blind to genotype and treatment. In all behavioural training and testing sessions, mice were allowed to acclimate to the testing room for 1 h. Additional details on specific tests and kinematic analysis are provided in the Appendix—Material and Methods.

## Statistical analysis

All statistical analyses for behaviour tests in R6/2 mice were performed using a linear mixed effect regression model with 95% confidence intervals calculated at each time point. Analysis of normality for all data sets was performed using the D'Agostino–Pearson omnibus test. One-way ANOVA followed by Tukey's multiple comparison test was used to compare brain weight at different ages in R6/2 mice and for the analysis of pDARPP32 and DARPP32 in Q140 mice. One-way ANOVA followed by Bonferroni post-tests was used in behavioural experiments involving Q140 mice. Two-tailed *t*-test was used when two groups were compared to each other. Two-way ANOVA followed by Bonferroni or Holm–Sidak post-test was used for comparisons including two genotypes and two treatments. The chi-square test was used to determine whether arm alternation by mice in the Y-maze was different than expected by chance. The non-parametric Mann–Whitney test was used for analysis of HTT aggregates, where a normal distribution of data could not be demonstrated. For kinematics data, separate three-factor ANOVAs (GROUP [3] × SEX [2] × PUMP [2]) were used to

**The paper explained**

**Problem**
Huntington's disease (HD) is a neurodegenerative disorder character-ized by motor, psychiatric and cognitive problems. There is no cure for HD and treatments that slow down disease progression while attenuating all disease symptoms are urgently needed.

**Results**
Our studies show that a molecule called ganglioside GM1 is able to ameliorate all aspects of HD, from motor symptoms to cognitive prob-lems and anxiety and depression, when it is administered chronically into the brain of animal models of the disease. The effects of GM1 extend beyond the treatment of HD symptoms. GM1 administration results in decreased levels of the disease-causing protein, mutant huntingtin (mHTT), which is a primary target in HD therapy. This is accompanied by decreased neurodegeneration along with several other beneficial effects, including normalization of levels of key neuro-transmitters in the brain and restoration of markers of striatal neuron, astrocyte and microglia functions.

**Impact**
The profound and widespread therapeutic effects of GM1 reported in this paper demonstrate that GM1 is a novel HD disease-modifying treatment in mice that warrants clinical investigations in HD patients.

determine any effects of the sex (SEX), the side at which the pump hang (PUMP) and the treatment group (GROUP) on the different kinematic measures. All comparisons were performed using a statis-tical significance level of 0.05, and all *P*-values were corrected for multiple comparisons.

**Expanded View** for this article is available online.

## Acknowledgements
This work was supported by grants from the Canadian Institutes of Health Research (CIHR, MOP 111219 and PDD 115217), from Alberta Innovates Health Solutions/Pfizer and from the Huntington Society of Canada to SS. MA was supported by a CIHR studentship. SS was supported by a Canada Research Chair (Tier 2).

## Author contributions
MA and DG designed and performed experiments, analysed data and wrote the manuscript; JF, LCM and ADP performed experiments and analysed data; PK and SWL performed experiments; AH, BJK, GBB, KF and KGT provided meth-ods and supervised experiments and data analysis; SS designed research and experiments, supervised experiments and data analysis and wrote the manuscript.

## Conflict of interest
SS holds a non-provisional US patent for the use of GM1 in treating HD. SS and the other authors declare no other financial interests or potential conflict of interest.

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
