## [Review Process File · EMBO Molecular Medicine]

Disease-Modifying Effects Of Ganglioside GM1 In Huntington's Disease Models

Melanie Alpaugh, Danny Galleguillos, Juan Forero, Luis Carlos Morales, Sebastian W. Lackey, Preeti Kar, Alba Di Pardo, Andrew Holt, Bradley J. Kerr, Kathryn G. Todd, Glen B. Baker, Karim Fouad and Simonetta Sipione

Corresponding author: Simonetta Sipione, University of Alberta

Review timeline:

Submission date:	01 March 2017
Editorial Decision:	27 April 2017
Revision received:	13 July 2017
Editorial Decision:	24 August 2017
Revision received:	06 September 2017
Accepted:	08 September 2017

Transaction Report:

Editor: Céline Carret

1st Editorial Decision

27 April 2017

Thank you for the submission of your manuscripts to EMBO Molecular Medicine. We have now heard back from the three referees whom we asked to evaluate your manuscript.

The referees have evaluated the two articles (EMM-2017-07763 and EMM-2017-07764) as back-to-back, therefore while their comments are specific to each article as you will see below, my editorial evaluation is common to both:

Overall, all referees find the topic of interest and are supportive of publication. Details have to be provided, along with better presentation of the data and statistical analysis (ref.3). Suggestions have been made to further improve the conclusions and we believe these would make the study more compelling.

One critical point common to all three referees however, and shared by the editorial office, is that both articles should be combined and you will see that referee 2 suggests a nice way forward to do this (please be reminded that you can accommodate up to 5 expanded view figures that would display online along with regular figures). We note that referee 3 highlights discrepancies in the experimental design that may complicate the combination of the two parts of the data. Nevertheless, we would like to strongly encourage you to do so, as we agree with the referees that on its own one article suffers from limited novelty and mechanism and the other one sounds rather preliminary. I hope you will be amenable to this option.

Revised manuscripts should be submitted within three months of a request for revision; they will

otherwise be treated as new submissions, except under exceptional circumstances in which a short extension is obtained from the editor.

Please note that EMBO Molecular Medicine policy encourages a single round of revision and that, as acceptance or rejection of the manuscript will depend on another round of review, your responses should be as complete as possible.

I look forward to receiving your revised manuscript.

***** Reviewer's comments EMM-2017-07763 *****

Referee #1 (Remarks):

The manuscript by Alpaugh, Galleguillos et al describe the preclinical assessment of the ganglioside GM1 treatment on mouse models of Huntington's disease.

The manuscript is well written and the data presented is strong. However, the data is mainly confirmative of a previous study from the same group using the YAC128 model (Di Pardo et al, PNAS). The new data clearly further supports the potential neuroprotection of GM1 treatment on different mouse models. However, overall, the new findings do not provide sufficient mechanistic insights into the neuroprotective role of GM1 treatment in HD.

The authors present an accompanied manuscript describing the effects of GM1 treatment in non-motor behaviours of different HD mouse models. In my view, a combination of data from both manuscripts is likely to provide enough novelty to warrant publication.

Referee #2 (Comments on Novelty/Model System):

The authors' study was sufficiently powered to detect the modest but significant deficits exhibited by the Q140 knock-in HD mouse model. Standard deviation error bars were used in all figures which allowed the reader to better judge the differences obtained following treatment with the GM1 ganglioside. The therapeutic strategy tested by the authors is capable of impacting both motor and non-motor HD mouse model phenotypes (in both a truncated and full-length mutant Huntingtin mouse model), and if further validated by additional pre-clinical and clinical trials, has the potential to become an important therapeutic strategy for HD.

Referee #2 (Remarks):

Previous work from the authors' laboratory had demonstrated that gangliosides are reduced in the brains of Huntington's disease (HD) mouse models and that intraventricular administration of GM1 ganglioside to the YAC128 HD mouse model can ameliorate its motor phenotypes. In this manuscript, the authors provide evidence that administration of a synthetic GM1 ganglioside to two additional HD mouse models (the R6/2 transgenic and Q140 knock-in) reduces levels of mutant Huntingtin (both soluble and aggregated), normalizes striatal DARPP32 levels (in Q140 heterozygous mice), reduces aberrant ferritin accumulation (in R6/2 mice), and normalizes both GFAP and Iba1 expression to wild type levels in R6/2 mice. Motor phenotypes (including gait

abnormalities) were also ameliorated in the Q140 mice, and significantly rescued in R6/2 mice. Based on these results (and the results obtained demonstrating cognitive improvements following GM1 ganglioside administration in an accompanying paper), the authors propose that GM1 ganglioside has the potential to be as effective as antisense oligonucleotide-mediated reduction of Huntingtin expression for the treatment for HD.

Strengths of this study include the validation of prior work using the YAC128 HD mouse model by using two additional models-R6/2 mice expressing an N-terminal fragment of mutant Huntingtin, and a knock-in mouse model expressing endogenous levels of full-length mutant Huntingtin. The experiments were also sufficiently powered to detect changes in the modest phenotypes exhibited by the 6-8-month-old Q140 mice. As the authors note, a particularly important finding in this study is that the GM1 ganglioside treatment affected the levels of both soluble and aggregated Q140 mutant Huntingtin without affecting wild type Huntingtin levels. Interestingly, in the R6/2 model, mutant Huntingtin aggregates were reduced in the cortex, but not striatum, following GM1 ganglioside administration.

The data provided in the manuscript support for the most part the authors' conclusions, and represents an important advance for the fields of HD and neurodegenerative research, but the manuscript can be potentially improved by addressing the following questions/comments:

1. Prior work by the authors has shown that GM1 ganglioside administration increases Huntingtin phosphorylation at S13 and S16. Was this also observed in the R6/2 and Q140 models after GM1 treatment? If so, this would provide additional data supporting a potential mechanism involved in GM1's ability to ameliorate a variety of HD mouse model phenotypes, including behavioral and cognitive deficits.
2. Is the time course of mutant Huntingtin aggregation in the R6/2 and Q140 models affected by GM1 ganglioside treatment? Based on the reduction of soluble mutant Huntingtin expression in the Q140 mice, this would be expected, but experimental verification would provide evidence that both phenotypic severity and potentially age-at-onset could be affected by GM1 ganglioside treatment.
3. The impact of this manuscript can be improved by incorporating results from the accompanying manuscript that show that GM1 ganglioside treatment significantly improves behavioral and cognitive deficits in the YAC128 and Q140 HD mouse models. Suggestions for how this could be accomplished are provided in the review of the accompanying manuscript.

Minor issues:

1. Figure S1: A legend for the different symbols used in the plot is needed.
2. Figure 1D: There appears to be an extraneous asterisk adjacent to the 10 wks GM1 bar.

Referee #3 (Comments on Novelty/Model System):

See review for authors.

Referee #3 (Remarks):

The potential use of GM1 gangliosides is an exciting and timely approach to treat Huntington's disease and the authors have used a number of models in the two manuscripts to assess mechanism and potential mechanisms involved. The authors have extended their previous work on GM1 in additional mouse models and included a number of new assays to assess both mechanism and efficacy. They have also evaluated expression of key dysregulated proteins in the mice and show restoration in their levels. The ferritin data are particularly striking. Further, the manuscript background and literature review is comprehensive, current, and well written. GM1 has extensive protective effects on numerous assays. The authors somewhat overstate the human impact, however the studies do suggest this could be an important avenue for therapeutic testing.

There is concern however relating to some general lack of detail about experimental design, time of administration, numbers of n's that seem to be variable, and age of mice at testing and the differences in ages of mice in the two manuscripts. In general would recommend that data should be presented as box plots. The error bars are large in the paper. This would better present the variability

in the data such as whether a single mouse is skewing the data or whether there is significant variability

Pooling data from different ages and cohorts can be an issue from a statistical rigor standpoint. Further, it is a bit unusual to pool data from het and homo Q140 mice, "where we did not detect differences in performance". This should be further clarified. Finally treatment is initiated in some cases in symptomatic mice and other treatments are begun at earlier time points. This is not discussed with respect to how one interprets the data or what it could mean in the context of HD treatment. The fact that GM1 has an effect after onset of symptoms is significant.

1. Figure 3 - need to show a representative image for the immuno to get a sense of the quality of the staining. Further the astrocyte data is confusing - in results the authors state that they are only referring to stats that show area, but it is not clear whether the area refers to counts or size of astrocyte cell body. Finally, potentially quantifying morphology could address question of decreased GFAP expression in spite of a higher number of GFAP+ astrocytes in cortical tissue.

2. For figure 3D, it would be helpful to clarify the variability in the n's used per group...For instance - there are 5 CSF WT for striatum but 4 for cortex and 2 HD GM1 for Striatum but 3 for cortex. The data is highly variable on the westerns and it would be helpful to have clarification of numbers. Finally, there are different ns in the data for GFAP and Iba1 - seems as though they should be same samples, but numbers of n are different.

3. There is no Q7/7 GM1 treatment group to assess effects on non-HD mice

4. Figure 5 - a loading control for HTT is needed on the blot? Further for the filter blots: R6/2 striatum and Q140 cortex seems to be missing - those could be included or use of specific tissue clarified - not clear why different tissues used for the two models.

5. The authors state that the R6/2 mice performed poorly in RR task. The WT mice did poorly also, which potentially reflects poor study design. Would suggest not including this data or something unusual with the cohort of mice.

6. For Q140 RR the authors state they only used females because males did not display impaired behavior. However, given that the exact age of mice is not clear, it is difficult to interpret these data. It might be helpful to use information from Hickey et al "Improvement of neuropathology and transcriptional deficits in CAG 140 knock-in mice supports a beneficial effect of dietary curcumin in Huntington's disease. *Mol Neurodegener.* 2012; 7: 12." for data that shows Q140 RR behavior and claims there is no difference between males and females. These authors show ways to show data where there is a treatment effect.

***** Reviewer's comments EMM-2017-07764 *****

Referee #1 (Remarks):

The manuscript by Alpaugh et al describes the preclinical assessment of the ganglioside GM1 on non-motor behaviours of mouse models of Huntington's disease.

The manuscript is well written and the data presented is strong. It complements previous data from the same group focusing on motor behaviour and huntingtin phosphorylation using the YAC128 model (Di Pardo et al, PNAS). The new data presented here focuses on non-motor behaviours and neurochemistry and provides further supports to the usage of GM1 as a possible HD treatment using different mouse models. The data presented is comprehensive, testing different aspects of non-motor phenotypes on a panel of HD mouse models. The big effort on using a panel of different tests on a number of HD mouse models is commendable. However, despite the amount of data presented, on its own the manuscript still lacks the novel insights required for publication.

The authors present an accompanied manuscript describing the disease modifying effects of GM1 treatment using different HD mouse models. In my view, a combination of data from both manuscripts is likely to provide enough novelty to warrant publication.

Referee #2 (Comments on Novelty/Model System):

Statistical analyses were sufficiently powered and performed appropriately. Demonstration that GM1 ganglioside administration can ameliorate or significantly improve no motor phenotypes in three HD mouse models (R6/2, YAC128, and Q140), in addition to ameliorating motor phenotypes (accompanying paper) represents a significant advance for the field and for developing therapeutics for the treatment of neurodegenerative disease. If validated by further pre-clinical experiments and clinical trials, G1 administration may represent a feasible means to treat HD and potentially, other neurodegenerative diseases.

Referee #2 (Remarks):

Prior work from the authors' laboratory and the results presented in the accompanying manuscript: "Disease-Modifying Effects of Ganglioside GM1 in Huntington's Disease Models" demonstrate that intraventricular administration of GM1 ganglioside can rescue or significantly improve YAC128, R6/2, and Q140 HD mouse model motor and neuropathological phenotypes. In this manuscript, the authors provide data showing that GM1 ganglioside treatment also ameliorates cognitive deficits and behavioral phenotypes that model depression and anxiety in the HD mice. Moreover, the rescue of behavioral and cognitive phenotypes is accompanied by normalization of HD model neurochemical changes in the brains of the mice.

Strengths of this study include the use to two full-length HD mouse models, inclusion of both males and females in the analyses, and incorporation of a variety of cognitive and behavioral tests that assess phenotypes corresponding to the behavioral/psychiatric symptoms observed in HD patients. However, the significance of the authors' findings would likely be enhanced if the data in this manuscript were included with the results of the motor and neuropathological characterizations presented in the accompanying manuscript. Combining the data into one manuscript would help to emphasize that GM1 ganglioside treatment is able to rescue motor, cognitive, behavioral, and neuropathological phenotypes in several HD mouse models. Overall, the data is convincing, the manuscript is an important contribution to the field, and the authors' findings should also be of interest to those developing therapeutics for other neurodegenerative diseases.

Suggestions for a combined manuscript:

From the accompanying manuscript, retain-

Figure 1. GM1 slows disease course in R6/2 mice.

Figure 5. Mutant HTT protein levels are reduced by administration of GM1.

Figure 6. GM1 improves motor behavior in R6/2 and Q140 mice.

Figure 7. GM1 corrects gait abnormalities in Q140 mice.

(Figures 2, 3, 4 can be included in the Supplementary Material.)

From this manuscript, retain-

Table 1.

Combine Figures 1 and 2 into one Figure. (e.g. GM1 normalizes anxiety- and depression-like behavior in HD mouse models.)

Combine Figures 3, 4, and 5 into one Figure. (e.g. GM1 improves nesting behavior, social cognition/memory, and open field habituation in YAC128 mice.)

Figure 6. GM1 improves the performance of heterozygous Q140 mice in the Y-maze.

Figure 7. GM1 restores wild-type levels of neuroactive amino acid levels in the cortex of YAC128 mice.

As the authors speculate in their discussion, the pleiotropic effects of GM1 ganglioside on HD mouse model phenotypes is likely related to a reduction in mutant Huntingtin levels, restoration of ganglioside deficits, and potentially to additional positive effects of GM1 ganglioside administration on NT signaling, neurogenesis, and calcium homeostasis. The authors also suggest that because GM1 ganglioside does not penetrate the BBB, intrathecal administration may be required. If so, the authors may want to discuss the feasibility of future preclinical studies that combine both an anti-HTT ASO and GM1 ganglioside administration via intrathecal injection.

Minor issues:

1. In the Figure and Supplementary Figure legends, standard deviation is abbreviated as STDEV,

- while in the accompanying manuscript, it is abbreviated as SD.
2. Figure S3. "Bars are means {plus minus}SD" should be added to the Figure legend
 3. Discussion, p. 16: intrathecal is misspelled.

Referee #3 (Comments on Novelty/Model System):

See review.

Referee #3 (Remarks):

The potential use of GM1 gangliosides is an exciting and timely approach to treat Huntington's disease and the authors have used a number of models in the two manuscripts to assess mechanism and potential mechanisms involved. The authors have extended their previous work on GM1 in additional mouse models and included a number of new assays to assess both mechanism and efficacy. In this manuscript, the authors have done extensive behavioral analysis, including novel assays, which are established and published, to assess psychiatric features of disease in HD mice. This is a significant benefit to the field to have this data. Further, GM1 has extensive protective effects on numerous assays. The authors somewhat overstate the human impact, however the studies do suggest this could be an important avenue for therapeutic testing.

There are concerns relating to lack of detail regarding experimental design, time of administration and age of Q140 mice at testing - this needs to be clarified. It is confusing when the various studies were performed and there is a disconnect in ages of mice for the two papers. For instance - "Treatment was performed for 28 days in R6/2 mice, and for 28 or 42 days in Q140 mice, starting at 6-8 months of age for Q140 mice, and at 6-8 weeks of age for R6/2 mice. Several cohorts of mice were used for the experiments. For each experimental cohort, each behavioral test was performed on the same day from the beginning of treatment. Data from all cohorts were pooled and analyzed together."

1. There is an issue with pooling heterozygous and homozygous Q140 data with an unclear study design for "GM1 treatment was performed at 6-6.5 months or 9-10 months of age in YAC128 mice and WT littermates; at 6-8 months of age in Q140 and Q7/7 mice". Some additional statistical analysis justifying this pooling would need to be provided
2. There is no Q7/7 GM1 treatment group which would be important for assessing GM1 effects on non-HD mice.
3. The fecal boli data is the only data in the manuscript for R6/2 mice and seems out of place.
4. Page 6 "Differently from imipramine, however, GM1 required to be administered for longer than seven days to improve depression-like behaviour in YAC128 mice (data not shown)." -Data should be shown in supplement.
5. p. 7. The nest-building assay was not significant in Q140 mice; do not need to report "trend"
6. There is a restoration of various neuroactive amino acid levels, however error bars are large and presenting the data as box plots might assist in interpretation.

1st Revision - authors' response

13 July 2017

EMM-2017-07763 (Original title: Disease-modifying effects of ganglioside GM1 in Huntington's disease models)

RESPONSE TO REVIEWER #1

1.1 Comment: *In my view, a combination of data from both manuscripts is likely to provide enough novelty to warrant publication.*

Response: We have combined the two manuscripts into one as suggested. The revised merged manuscript now includes 8 figures, 1 table and 2 expanded view figures, according to EMBO Molecular Medicine guidelines:

Fig. 1 GM1 decreases neuropathology and weight loss in R6/2 mice
Fig. 2 Effects of GM1 on astroglial and microglial markers.
Fig. 3 Mutant HTT protein levels are reduced by administration of GM1
Fig. 4 GM1 improves motor behavior in R6/2 and Q140 mice
Fig. 5 GM1 corrects gait abnormalities in Q140 mice
Fig. 6 GM1 improves anxiety-like and depression-like behavior in HD mice.
Fig. 7 GM1 improves the performance of HD mice in social and cognitive tests
Fig. 8 GM1 restores wild-type levels of neuroactive amino acid levels in the cortex of YAC128 mice.
Table 1 Effect of GM1 treatment on biogenic amines
Fig. EV1 GM1 increases DARPP32 and its phosphorylation in Q7/Q140 mice
Fig. EV2 GM1 decreases depression-like behavior in Q140 mice and in older WT mice.
 In addition, the Appendix includes 6 supplementary figures and 1 supplementary table.

RESPONSE TO REVIEWER #2

2.1 Comment: *Prior work by the authors has shown that GM1 administration increases huntingtin phosphorylation at S13 and S16. Was this also observed in the R6/2 and Q140 models after GM1 treatment?*

Response: Unfortunately we were unable to measure S13 and S16 phosphorylation in this new study because the phospho-specific antibodies used in our previous work were no longer available.

2.2 Comment: *Is the time course of mutant huntingtin aggregation in the R6/2 and Q140 models affected by GM1 treatment? Based on the reduction of soluble mutant huntingtin expression in the Q140 mice, this would be expected, but experimental verification would provide evidence that both phenotypic severity and potentially age-at-onset could be affected by GM1 treatment.*

Response: A time-course of the effects of GM1 on mutant huntingtin aggregation was not performed. We believe that end-of-treatment measurements are sufficiently informative to allow us to make predictions on the disease-modifying activity of the drug, while at the same time keeping the number of animals used in the experiments within a practical range. Although we have not directly shown whether GM1 could affect age at onset, our data demonstrate nevertheless that GM1 slows down progression (and therefore potentially also age at onset), based on its effects on soluble and insoluble mutant huntingtin and on the reduced neurodegeneration measured in R6/2 mice.

2.3 Comment: *The impact of the manuscript can be improved by incorporating results from the accompanying manuscript.*

Response: We have incorporated those data as suggested. See response to comment 1.1 for a summary of the figures in the revised merged manuscript.

2.4 Response to Minor issues:

Fig. S1: A legend for the different symbols used in the plot is needed.

Symbols are included in the revised manuscript (now Fig. S2)

Fig 1D: There appears to be an extraneous asterisk adjacent to the 10wks GM1 bar. The asterisk has been removed.

RESPONSE TO REVIEWER #3

3.1 Comment: *There is concern relating to some general lack of detail about experimental design, time of administration, numbers of n's that seem to be variable and age of mice at testing and the differences in ages of mice in the two manuscripts.*

Response: We believe we have now clarified experimental details in the material and methods and in the supplementary material and methods of the revised manuscript. In addition, we have added a supplementary figure (Fig. S1 in the revised manuscript) that illustrates the experimental design, including animal models used, age of the mice at the beginning of treatment, duration of treatment

and tests and analysis performed. In Fig. S1B we present the timeline of experiments in the three different animal models used.

The number of animals in each test is indicated in the figure legends. Ns are sometime different in different tests, as a result of different cohorts of mice being tested, or if individual animals were excluded from testing or from data analysis. When this was done, exclusion criteria were applied in compliance with the ARRIVE guidelines. Exclusion criteria are reported in the Appendix Material and Methods of the revised manuscript.

All mice used in our study were already symptomatic (as determined by motor tests) at the time of treatment. R6/2 mice were used at a much younger age than Q140 and YAC128 mice because they develop an early phenotype and have significantly shortened lifespan. In the case of YAC128 mice we used two cohorts of mice, one at 6 months of age and the second at 9 months of age, for practical reasons, as we could not get a group of animals of the same age large enough for our study. Both groups were fully symptomatic and each group was used for different tests, as indicated in Fig. S1A of the revised manuscript, except for the forced swim test, which was repeated in both groups (data presented independently for each age group in Fig. 6H and Fig. EV2). The inclusion of a second cohort of older mice (9 months at the beginning of treatment) shows that GM1 exerts therapeutic effects even in older mice and as disease progresses.

3.2 Comment: *In general, would recommend that data should be presented as box plots.*

Response: We have changed our figure style to box plots as suggested, with the exception of a few cases where individual datapoints are presented, when one or more experimental groups contained less than 5 datapoints (according to EMBO guidelines).

3.3 Comment: *Pooling data from different ages and cohorts can be an issue from a statistical rigor standpoint.*

Response: Due to the complexity of the experimental design, the large number of animals used, and the large variety of tests and measurements performed - which could not have been performed all at once - we had to use various cohorts of mice. Please note that each cohort contained the same number of experimental groups, which were carefully balanced for genotype, sex, age and weight. Furthermore, animal testing was performed according to a rigid schedule, with the same tests performed on the same day after beginning of treatment and at the same time of the day across multiple cohorts. We would like to argue that this is potentially a strength of our study, rather than a weakness, because it demonstrates that the effects of GM1 were independent from potential environmental factors beyond the control of the experimenter - from changes in animal facility personnel to exposure to noise or other potential forms of stress occurring even in the most rigorously controlled animal facilities - that could affect any mouse colony at any given time.

3.4 Comment: *It is a bit unusual to pool data from het and homo Q140 mice. This should be further clarified.*

Response: To the best of our knowledge and based on our data, heterozygous and homozygous Q140 mice generally show similar disease/phenotype severity. This justifies pooling them together to increase the N in our study. In the revised manuscript we have included a table (Appendix Table S1) that includes statistical analysis to demonstrate that the behavior of the two genotypes was similar where data were pooled.

3.5 Comment: *Treatment is initiated in some cases in symptomatic mice and other treatments are begun at earlier time points. This is not discussed with respect to how one interprets the data or what it could mean in the context of HD treatment.*

Response: All treatments were initiated in already symptomatic mice (as determined by motor impairment). This is now stated in the Material and Methods of the revised manuscript.

3.6 Comment: *Fig.3 – need to show a representative image for the immune to get a sense of the quality of the staining. It is not clear whether the area refers to counts or size of astrocyte cell body. Potentially quantifying morphology could address question of decreased GFAP expression in spite*

off a higher number of GFAP+ astrocytes in cortical tissue.

Response: We have added representative images of the immunostaining with anti-GFAP and anti-Iba1 antibodies (Fig. 2 in the revised manuscript). Data in Fig. 2 show the percent of the total area of each photomicrograph immunoreactive to anti-GFAP antibodies, after applying a threshold in ImageJ to eliminate background signal. This method of analysis provides a global measurement that does not discriminate between the number of astrocytes and size of cell bodies, but is reflective of both. Thus, this method would detect changes in the number and/or in the extension of individual astrocytes. A detailed analysis of astroglia morphology, although potentially interesting, was beyond the scope of this work, which was focused on demonstrating more general diseasemodifying properties of GM1, and therefore it was not performed.

3.7 Comment: *For Fig. 3D, it would be helpful to clarify the variability in the n's used per group....For instance – there are 5 CSF WT for striatum but 4 for cortex and 2 HD GM1 for striatum, but 3 for cortex. The data is highly variable on the westerns and it would be helpful to have clarification of numbers.*

Response: We apologize for the confusion generated. The blot in Fig. 3 (now Fig. 2 in the revised manuscript) is representative of only a subset of tissue lysates analyzed by immunoblotting, while the graph in the same figure shows the average of 6-7 animals per group, run in different gels, with each gel including all experimental groups. The exact N is indicated in the figure legend. Concerning histological analysis, the number of animals used depended on the availability of brain coronal sections spanning the region of interest. From each brain we obtained thirteen 20 µm-thick serial sections. For consistency and reproducibility, each immunohistochemical marker was analyzed in the same brain area, and therefore in the same serial sections (± 1 section) for each animal. In a few occasions the staining for a specific marker had to be repeated for technical reasons. When serial sections corresponding to the same brain area were no longer available, that specific animal had to be excluded from the analysis, thus generating an unequal N for the various experimental groups. When staining had to be repeated for technical reasons, and if sections of the same region were no longer available the animal would have to be excluded from the analysis. This is why fewer mice were analysed for Iba1 than for GFAP.

3.8 Comment: *Fig. 5 – A loading control for HTT is needed on the blot?*

Response: In Fig. 5 (now Fig.3 in the revised manuscript), in order to control for consistent sample loading we used Swift™ staining, a total protein stain widely used to normalize protein content in immunoblots. This is now clarified in the figure legend in the revised manuscript.

3.10 Comment: *The authors state that the R6/2 mice performed poorly in RR task. The WT mice did poorly also, which potentially reflects poor study design. Would suggest not including these data or something unusual with the cohort of mice.*

Response: We have removed the rotarod task for the R6/2 mice in the revised manuscript, as suggested.

3.11 Comment: *For Q140 RR, the authors state they only used females because males did not display impaired behavior. However, given that the exact age of mice is not clear, it is difficult to interpret data. It might be helpful to use information from Hickey et al. for data that show Q140 RR behavior and claims there is no difference between males and females. These authors show ways to show data where there is a treatment effect.*

Response: The apparent discrepancy between our data concerning the performance of Q140 mice in the rotarod and data presented by Hickey et al. might originate from the fact that younger mice (4.5 months) were used in Hickey et al., while our Q140 mice were 7.5 month-old at the time of testing (with treatment started at 6.4 months). We have added a comment in the revised manuscript to address this point and suggested that female Q140 mice might develop a progressive motor impairment on the rotarod test as they age. Concerning sex-related differences in the rotarod performance, we have added statistical data to support our claim in the revised document, on page 10 (effect of sex: $F_{1,56}=6.7, p=0.01$).

EMM-2017-07764 (Original title: Ganglioside GM1 restores normal non-motor behavior in Huntington's disease mouse models)**RESPONSE TO REVIEWER #1.**

1.1 Comment: *In my view, a combination of data from both manuscripts is likely to provide enough novelty to warrant publication.*

Response: We have combined the two manuscripts into one as suggested. See above for details.

RESPONSE TO REVIEWER #2.

2.1 Comment: *The Authors may want to discuss the feasibility of future preclinical studies that combine both an anti-HTT ASO and GM1 ganglioside administration via intrathecal injection.*

Response: We have added a comment on the potential for combined therapies in the discussion of the paper.

2.2 Response to minor issues:

1) Standard deviation is now abbreviated as SD. Most graphs, however, now show box-and whisker plots with median, minimum and maximum values.

2) We were asked to provide a description for what bars represent in the legend of Figure S3. This is now Fig. EV2 in the revised manuscript. Data are now presented as box-plots.

3) We have corrected the word “intrathecal” in the discussion.

RESPONSE TO REVIEWER #3

3.1 Comment: *There are concerns relating to lack of detail regarding experimental design, time of administration and age of Q140 mice at testing – This needs to be clarified.*

Response: We believe we have clarified the confusion in the merged manuscript. See response 3.1 above for details.

3.2 Comment: *There is an issue with pooling heterozygous and homozygous Q140 data with an unclear study design. Some additional statistical analysis justifying this pooling would need to be provided.*

Response: In the revised manuscript we have included a table (Appendix Table S1) that includes statistical analysis to demonstrate that the behavior of the two genotypes was similar in most tests and to justify pooling of data from these two genotypes.

3.3 Comment: *The fecal boli data in the only data in the manuscript for R6/2 mice and seems out of place.*

Response: In the revised merged manuscript inclusion of R6/2 data for the fecal boli test is now fully justified.

3.4 Comment: *Differently from imipramine, GM1 required to be administered for longer than seven days to improve depression-like behavior in YAC128 mice. Data should be shown in supplement.*

Response: The data are now presented in the Appendix Fig. S6.

3.5 Comment: *The nest-building assay was not significant in Q140 mice; do not need to report “trend”.*

Response: This test is no longer presented in the revised manuscript.

3.6 Comment: *There is a restoration of various neuroactive amino acid levels, however error bars are large and presenting the data as box plots might assist interpretation.*

Response: Data are now presented as box plots.

2nd Editorial Decision

24 August 2017

Thank you for the submission of your revised manuscript to EMBO Molecular Medicine. We have now received the enclosed reports from the referees that were asked to re-assess it. As you will see the reviewers are now supportive and I am pleased to inform you that we will be able to accept your manuscript pending the following final amendments:

- 1) Please address referee 1 comments in a point-by-point rebuttal letter as well as discuss accordingly in the discussion section of the article.
- 2) In legends exact n= and exact p= values, not a range. Some people found that to keep the figures clear, providing a supplemental table with all exact p-values was preferable. You are welcome to do this if you want to.

Please submit your revised manuscript within two weeks. I look forward to seeing a revised form of your manuscript as soon as possible.

***** Reviewer's comments *****

Referee #1 (Remarks):

The manuscript now combines data from the manuscripts previously submitted.

The result is a much improved manuscript with wide ranging data showing that ganglioside treatment is beneficial in different HD mouse models, positively affecting a range of areas including: huntingtin levels, brain/striatal atrophy, motor and cognitive deficits as well as different neurochemical levels.

Overall, the paper is now suitable for publication.

The only minor comment I have would be to perhaps include in the discussion the possibility that GM1 may affect autophagy and thus affect the clearance of mutant Htt, potentially contributing to the underlying beneficial effects of GM1 treatment. There is some conflicting evidence in the literature regarding gangliosides and their possible effects on autophagic flux, and I think it would improve the manuscript to at least discuss this possibility. At this stage, I won't suggest the authors to measure autophagosomes via LC3-II levels, but this could be a possibility they may consider for the discussion and/or for the future.

Referee #2 (Comments on Novelty/Model System):

The experiments were sufficiently powered to detect the effects of ganglioside GM1 administration on the phenotypes exhibited by three HD mouse models (Q140, R6/2, and YAC128). The authors demonstrate that Intraventricular GM1 administration can ameliorate or significantly modify mutant Htt levels, mutant Htt aggregation, and both motor and cognitive phenotypes in the HD mouse models.

Referee #2 (Remarks):

The authors have successfully combined their two previous manuscripts describing the disease-modifying effects of GM1 administration on HD mouse model phenotypes into a single manuscript that is now better organized and provides a more compelling story about the potential for GM1 administration as a therapy for HD. In addition, the authors have adequately addressed all my questions and concerns about their original manuscripts in their revised manuscript.

Referee #3 (Remarks):

The authors have revised the manuscript as suggested by the reviewers and have addressed concerns sufficiently. Like the combined version of the manuscript much better.

2nd Revision - authors' response

06 September 2017

Response to Referee #1.

Comment: *The only minor comment I have would be to perhaps include in the discussion the possibility that GM1 may affect autophagy and thus affect the clearance of mutant HTT, potentially contributing to the underlying beneficial effects of GM1 treatment. There is some conflicting evidence in the literature regarding gangliosides and their possible effects on autophagic flux, and I think it would improve the manuscript to at least discuss this possibility.*

Response: We thank the reviewer for his/her suggestion. Indeed we intend to test the potential effects of GM1 on autophagy as a mechanism to reduce HTT levels in our models, and we have discussed this possibility in the revised manuscript.

Corresponding Author Name: Simonetta Spione

Manuscript Number: EMM-2017-07763